# RADE: Random Add-Drop Edge as a Regularizer

**Danial Saber** [1]  **Amirali Salehi-Abari** [1]

## Abstract

Graph Neural Networks (GNNs) suffer from overfitting and over-squashing of long-range information. Stochastic graph augmentations (e.g., edge deletion) regularize training against overfitting but can introduce train-inference misalignment and do not improve over-squashing. In contrast, rewiring methods improve connectivity to mitigate over-squashing, but are not designed to regularize training. We propose *Random Add-Drop Edge (RADE)*, a stochastic graph augmentation method that jointly drops and adds edges to address both overfitting and over-squashing simultaneously. RADE is provably designed to align training and inference so that random augmentations regularize training without distribution shift, while supporting long-range communication at inference. We further propose and study a minibatch gradient-norm balancing algorithm that adapts deletion and addition rates during training, rendering RADE hyperparameter-free in practice. Experiments on node- and graph-classification benchmarks show that RADE is a strong regularizer and mitigates over-squashing. Ablations support the roles of train-inference alignment, adaptive rate selection, and the complementary effects of random edge deletion and edge addition.

## 1. Introduction

Graph Neural Networks (GNNs) (Gori et al., 2005; Micheli, 2009; Scarselli et al., 2008) are neural networks designed for relational data (e.g., social networks, protein-protein networks, etc.), with diverse applications such as recommender systems (Wu et al., 2022), knowledge graph completion (Arora, 2020), and molecular property prediction (Kearnes et al., 2016). The dominant paradigm of GNNs is Message-Passing Neural Networks (MPNNs) (Gilmer et al., 2017), in which each node iteratively aggregates and transforms messages from its neighbors. Despite GNNs' success, their generalizability faces several challenges, including limited expressiveness (Bevilacqua et al., 2022), over-smoothing (Li et al., 2019), over-squashing (Alon & Yahav, 2021), and overfitting. We focus on two problems of *overfitting*, a widespread challenge in machine learning, and *over-squashing*, a distinctive challenge for MPNNs. Over-squashing occurs when the set of nodes that influence a node's representation through message passing expands rapidly as the number of layers increases, forcing a lot of information to be compressed into a fixed-size embedding (Alon & Yahav, 2021). This limits the ability to capture higher-order and long-range dependencies in graphs.

Prior work addresses overfitting and over-squashing largely in separation. To reduce overfitting in GNNs, common approaches include penalty-based regularization (e.g., weight decay), architectural regularization (e.g., Dropout (Srivastava et al., 2014)), and stochastic graph augmentation as an implicit regularizer—mostly in the form of deletion-based perturbations (e.g., DropEdge (Rong et al., 2020)). Of special interest for GNNs is graph augmentation, but it is often tuning-sensitive: weak perturbations have little effect, while strong ones remove useful graph signal. Also, training on perturbed views can create train-inference misalignment unless an explicit correction is applied. To mitigate over-squashing, existing methods typically increase connectivity through rewiring or shortcut edges (Abboud et al., 2022; Arnaiz-Rodríguez et al., 2022; Karhadkar et al., 2023; Alon & Yahav, 2021). These methods can improve long-range information flow, but the extent of rewiring must still be tuned, and simple random edge addition is less emphasized than more structured surgical rewiring.

Looking across these two bodies of work, stochastic augmentations regularize training without improving long-range communication, whereas rewiring methods—viewed as a form of augmentation—improve connectivity without regularizing. This gap calls for a unified augmentation framework that regularizes training and, when needed, improves long-range connectivity against over-squashing. Since random edge addition has the potential to be a stochastic regularizer and to facilitate long-range connectivity, a natural question arises: can random edge addition be combined with edge deletion so that the same augmentation both regularizes training and, when useful, improves long-

---

[1]Ontario Tech University, Oshawa, Ontario, Canada. Correspondence to: Danial Saber <danial.saber@ontariotechu.ca>.

*Proceedings of the 43rd International Conference on Machine Learning*, Seoul, South Korea. PMLR 306, 2026. Copyright 2026 by the author(s).

range communication? The main difficulty is train-inference alignment, since training on randomly perturbed graphs but inferring on a fixed graph can induce misalignment or distribution shift. A second practical challenge is controlling the deletion and addition rates without dataset-specific tuning.

We propose *Random Add-Drop Edge (RADE)*, a stochastic topology augmentation that randomly drops existing edges and adds non-edges during training, with corrections for train-inference alignment. We introduce two variants of RADE. *RADE-OF* solely targets overfitting: it corrects both deletion and addition effects, so that training-time message aggregation, in expectation, matches the aggregation of the input graph at inference. Thus, the random perturbations act as training-time regularization without systematically shifting the inference rule. *RADE-OFS* targets both overfitting and over-squashing: it corrects the deletion effect but keeps the expected contribution of added edges at inference, thus preserving the regularization benefit of stochastic perturbations while creating additional communication paths for long-range information flow. For both variants, we further show that the regularization effects of random deletion and addition are generally not interchangeable. We also introduce a mini-batch gradient-norm balancing rule that adapts the drop/add rates during training, thereby making RADE practically hyperparameter-free.

Our extensive evaluation across diverse node- and graph-level datasets and various backbone models (e.g., GCN, GIN, and GAT) confirms RADE's potential for both implicit regularization and over-squashing mitigation. RADE-OF consistently outperforms competitive regularizers while RADE-OFS is the most effective when long-range communication is important: it yields the largest gains on over-squashing-prone graph classification benchmarks (Saber & Salehi-Abari, 2025) and datasets requiring long-range interactions (e.g., peptides-func) (Dwivedi et al., 2022). Ablations further verify the benefit of our train-inference alignment rules (i.e., removing alignment rules downgrades performance), the performance gains of our adaptive rate selection (i.e., adaptively selected drop/add rates outperform fixed, hyperparameter-tuned choices), and the complementary effects of random drop and addition (i.e., RADE outperforms the drop-only or add-only variants). Overall, RADE appears to be a simple and effective augmentation framework for regularizing against overfitting and mitigating over-squashing.

## 2. Related Work

Data augmentation for GNNs perturbs the input graph through its topology (edges/nodes) or attributes (node/edge features). Many augmentations are stochastic regularizers against overfitting, while others address structural limitations of MPNNs (e.g., over-squashing). We review prior

work through two lenses: overfitting and over-squashing.

**Overfitting.** Stochastic augmentations act as implicit regularizers against overfitting. Examples include random edge deletion (Rong et al., 2020), feature/representation noise injection (Veličković et al., 2019; Feng et al., 2019; Yang et al., 2021), feature/message masking (Thakoor et al., 2022; Fang et al., 2023; You et al., 2021), and node dropping to generate alternative views (Feng et al., 2020; Saber & Salehi-Abari, 2024; You et al., 2020; Louis et al., 2023). These perturbations can also mitigate *over-smoothing* in deep GNNs by weakening or randomizing propagation pathways (Rong et al., 2020; Thakoor et al., 2022; Fang et al., 2023).

**Over-squashing.** To improve long-range information flow, many methods add or rewire connectivity to alleviate bottlenecks (Alon & Yahav, 2021). This includes virtual nodes (Cai et al., 2023; Southern et al., 2025), higher-order structures (Bodnar et al., 2021a;b), last-layer full connectivity (Alon & Yahav, 2021), diffusion/multi-hop rewiring (Abboud et al., 2022; Abu-El-Haija et al., 2019; Wang et al., 2021; Nikolentzos et al., 2020; Gasteiger et al., 2019; Brüel-Gabrielsson et al., 2022; Barbero et al., 2024; Gutteridge et al., 2023), fully-connected attention in graph transformers (Ying et al., 2021; Kreuzer et al., 2021; Rampášek et al., 2022), and targeted edge addition around bottlenecks (Topping et al., 2022; Karhadkar et al., 2023).

**Other objectives.** Augmentations have also been used to improve expressiveness by ensembling diverse substructure views (Papp et al., 2021; Bevilacqua et al., 2022), to support robustness/denoising under structural noise or adversaries (Wu et al., 2019b; Zhang & Zitnik, 2020; Entezari et al., 2020), and to enable scalable stochastic training via sampled neighborhoods (Hamilton et al., 2017; Chen et al., 2018b;a; Jacob et al., 2023; Zeng et al., 2020; Louis et al., 2022).

**Our work.** Our work bridges two separate uses of graph augmentation—regularization against overfitting and connectivity enrichment against over-squashing—through random edge addition. Existing regularizers mostly drop graph signals (e.g., nodes, edges, etc.), while random edge addition remains underexplored as a regularizer for overfitting. In contrast, over-squashing methods often add edges deterministically or in a bottleneck-driven manner, rather than through uniformly random edge addition. RADE combines the two views through random add-drop edge augmentation: deletion and addition jointly induce a regularizer, while addition can enrich long-range communication paths.

## 3. Preliminaries and Background

We consider an undirected graph $\mathcal{G} = (V, E)$ with $n$ nodes and $m$ edges, represented by an adjacency matrix $\mathbf{A} \in \{0, 1\}^{n \times n}$, where $A_{ij} = 1$ iff $(i, j) \in E$. Each node $i$ has a $d$-dimensional feature vector $\mathbf{x}_i$, stacked as $\mathbf{X} \in \mathbb{R}^{n \times d}$.

**Message-Passing Neural Networks (MPNNs).** Most GNNs are MPNNs, in which node representations are iteratively updated via messages from their neighbors. Let $\mathbf{H}^{(\ell)} \in \mathbb{R}^{n \times d_\ell}$ denote the representations at layer $\ell$, where each row vector $\mathbf{h}_i^{(\ell)}$ corresponds to node $i$'s representation. The initial representations are $\mathbf{h}_i^{(0)} = \mathbf{x}_i$. For each layer $\ell \leq L$, each node $i$'s representation is updated by

$$\mathbf{h}_i^{(\ell)} = \mathrm{Up}^{(\ell)}\Big(\mathbf{h}_i^{(\ell-1)}, \mathbf{a}_i^{(\mathcal{G},\ell)}\Big), \qquad (1)$$

where the update function $\mathrm{Up}^{(\ell)}$ uses the previous representation $\mathbf{h}_i^{(\ell-1)}$, and the aggregated neighbor message

$$\mathbf{a}_i^{(\mathcal{G},\ell)} = \mathrm{Agg}^{(\ell)}\Big(\Big\{\mathbf{m}_{ij}^{(\ell-1)} : A_{ij} = 1, \, j \neq i\Big\}\Big). \quad (2)$$

Here, $\mathrm{Agg}^{(\ell)}$ is a layer-$\ell$ aggregation operator (e.g., sum/mean or degree-normalized aggregation) over $\mathbf{m}_{ij}^{(\ell-1)}$, where $\mathbf{m}_{ij}^{(\ell-1)}$ denotes the message of node $j$ to node $i$ based on their $(\ell-1)$-layer representations. Stacking layers allows information from multi-hop neighborhoods to influence final node $i$'s representation $\mathbf{h}_i^{(L)}$. Many GNNs use a weighted-sum aggregation operator.

**Definition 3.1** (Weighted-sum aggregation). $\mathrm{Agg}^{(\ell)}$ is a *weighted-sum* aggregator if there exist weights $\{\alpha_{ij}^{(\mathcal{G})}\}_{j \neq i}$ such that, for each node $i$,

$$\mathbf{a}_i^{(\mathcal{G},\ell)} = \sum_{j \neq i} \alpha_{ij}^{\mathcal{G}} \, \mathbf{m}_{ij}^{(\ell-1)}, \qquad (3)$$

The coefficients $\alpha_{ij}^{\mathcal{G}}$ depend on graph structure (e.g., degrees) and/or representations (e.g., attention).[1]

For example, in GIN (Xu et al., 2019), $\alpha_{ij}^{\mathcal{G}} = A_{ij}$ (sum aggregation); in GAT (Veličković et al., 2018), $\alpha_{ij}^{\mathcal{G}} = A_{ij} B_{ij}$, with $B_{ij}$ being the j's attention to $i$; and in GCN (Kipf & Welling, 2017), $\alpha_{ij}^{\mathcal{G}} = A_{ij}/\sqrt{d_i d_j}$, with degrees $d_i$ and $d_j$.

**Data Augmentation in Graphs.** Graph augmentation samples a *perturbed view* $\mathcal{G}'$ of an input graph $\mathcal{G}$ by modifying its topology $\mathbf{A}$ or node attributes $\mathbf{X}$. We model an augmentation method as a conditional distribution $\mathcal{Q}(\cdot \mid \mathcal{G})$ over graphs with a sampled view $\mathcal{G}' \sim \mathcal{Q}(\cdot \mid \mathcal{G})$. This view-sampling perspective unifies two common *purposes* of augmentation: mitigating overfitting and over-squashing. When $\mathcal{G}'$ is sampled *stochastically during training* (e.g., once per epoch) but inference uses the input graph $\mathcal{G}$, augmentation serves as implicit regularization against overfitting. In contrast, when augmentation is used to alleviate over-squashing, one typically constructs a single augmented graph $\mathcal{G}'$—i.e., $\mathcal{Q}$ is degenerate at $\mathcal{G}'$—and uses that same graph for *both*

---

[1] For notational brevity, we write $\alpha_{ij}^{\mathcal{G}}$ in place of $\alpha_{ij}^{(\mathcal{G},l)}$, suppressing the layer dependence when unambiguous.

*training and inference.* Such augmentations typically add shortcut edges that shorten long-range message-passing paths and improve information flow. These two uses suggest a combined setting: one may resample perturbed views stochastically during training while using a fixed augmented graph at inference, so that augmentation helps address overfitting during training and improves connectivity to mitigate over-squashing at inference.

**Train-Inference Alignment in Data Augmentation.** Since graph augmentations perturb message passing, a central question is whether the aggregation used during training aligns with the one used at inference, as failing to do so can hinder generalization. The nature of this alignment depends on the purpose of augmentation.

When augmentation is used to mitigate overfitting, a train-inference mismatch arises: the model aggregates messages on epoch-specific perturbed graphs during training, but on the fixed input graph during inference. To avoid potential train-inference mismatch, the training-time aggregation should be *corrected* so that it aligns with the inference-time aggregation in expectation under the augmentation $\mathcal{Q}$.

**Definition 3.2** (Expectation-Preserving Aggregation). Let $\widetilde{\mathbf{a}}_i^{(\mathcal{G}',\ell)}$ be the output of the corrected training-time aggregation $\widetilde{\mathrm{Agg}}$ on the perturbed graph $\mathcal{G}'$ under $\mathcal{Q}$, and $\mathbf{a}_i^{(\mathcal{G},\ell)}$ be the output of the inference-time aggregation $\mathrm{Agg}$. The training-time aggregation is *expectation-preserving* if, for every layer $\ell$ and node $i$,

$$\mathbb{E}_{\mathcal{G}' \sim \mathcal{Q}(\cdot \mid \mathcal{G})}\Big[\widetilde{\mathbf{a}}_i^{(\mathcal{G}',\ell)} \mid \mathbf{H}^{(\ell-1)}\Big] = \mathbf{a}_i^{(\mathcal{G},\ell)}. \qquad (4)$$

When this condition holds, stochastic training introduces mean-zero aggregation noise around the inference-time aggregation rather than a systematic bias. Many common graph augmentations (Rong et al., 2020; Feng et al., 2020) do not yield expectation-preserving aggregation by default—e.g., simple edge deletion (Rong et al., 2020) changes the expected aggregated messages—so achieving expectation-preserving criterion typically requires correcting the training-time aggregation. Definition 3.2 is not only an alignment criterion; it also provides the principle for deriving the corrected aggregation rule $\widetilde{\mathrm{Agg}}$ from any aggregation rule $\mathrm{Agg}$ under stochastic augmentation.

In contrast, when the goal is to mitigate over-squashing, one typically uses the same augmented graph $\mathcal{G}'$ in *both training and inference*. In this regime, the same aggregation $\mathbf{a}_i^{(\mathcal{G}',\ell)}$ is used in both phases, so there is no train-inference aggregation mismatch to correct.

If one seeks to mitigate overfitting and over-squashing simultaneously, and the same augmentation is used in both training and inference, then both phases induce the same change in message aggregation without any train-inference mismatch. However, if training uses stochastic perturbed

views while inference uses a fixed aggregation rule, then the expected training-time aggregation should be aligned with the chosen inference-time aggregation. In this case, expectation preservation should be defined not necessarily with respect to the input-graph aggregation, but with respect to the desirable inference-time aggregation that preserves the desirable augmentation effect (e.g., shortcuts for long-range communication).

**Augmentation for Implicit Regularization.** Under some simplifying assumptions, stochastic graph augmentation with expectation-preserving aggregation can be interpreted as an implicit regularizer (Fang et al., 2023). Consider binary cross-entropy loss under a *linear* MPNN,[2] where $L_{\mathrm{BCE}}$ and $\tilde{L}_{\mathrm{BCE}}$ are the BCE losses on the input and perturbed graphs, respectively. Then the expected loss follows the variance-based regularization view (Fang et al., 2023):

$$\mathbb{E}[\tilde{L}_{\mathrm{BCE}}] \approx L_{\mathrm{BCE}} + \frac{1}{2}\sum_i z_i(1-z_i)\mathrm{Var}(\delta_i), \quad (5)$$

where $z_i = \sigma(s_i)$, with $\sigma$ the sigmoid function and $s_i$ the logit on the input graph, and $\delta_i$ is the induced perturbation noise around $s_i$.[3] Eq. (5) shows that stochastic perturbation is a variance-based regularizer: training favors parameters that produce low-variance predictions across perturbed views, as in Dropout (Srivastava et al., 2014), DropMessage (Fang et al., 2023), and DropEdge (Rong et al., 2020).

**DropEdge.** DropEdge (Rong et al., 2020) randomly deletes edges during each epoch of training, so messages are aggregated on a perturbed graph with a stochastic subset of the original edges. This stochastic perturbation acts as an implicit regularizer. However, expectation-preserving aggregation is not automatic: edge deletion changes the expected aggregation unless the training-time aggregation is *corrected*. For sum aggregation (e.g., GIN), this correction reduces to rescaling by a global factor. For general weighted-sum schemes, however, deletion perturbs the aggregation in a way that is not generally corrected by a single global rescaling. This motivates us to develop a principled framework for deriving expectation-preserving corrections for general weighted-sum aggregators.

**Deletion vs. Addition as Distinct Topology Perturbations.** Random edge deletion stochastically suppresses existing message pathways and is primarily used as an implicit regularizer against overfitting. In contrast, random edge addition, often neglected in the literature, creates new pathways that can improve long-range information flow and mitigate over-

squashing while still providing stochastic regularization during training. Thus, deletion acts on existing edges, whereas addition acts on non-edges, so their effects are generally not interchangeable, even when viewed purely as stochastic regularizers. This asymmetry motivates studying edge addition as a complementary perturbation primitive and, more broadly, *add-drop* augmentations that combine both mechanisms within a unified view-sampling framework.

## 4. Random Add-Drop Edge

We propose **R**andom **A**dd-**D**rop **E**dge (RADE), a graph augmentation strategy with an explicit aggregation correction rule, designed to mitigate overfitting and over-squashing. RADE has two components: (i) *perturbation*, specifying how a perturbed view $\mathcal{G}'$ is sampled and (ii) *aggregation-correction*, detailing how message aggregation is corrected to control train-inference aggregation alignment.

**Perturbation**. At each epoch of training, RADE samples a perturbed adjacency $\mathbf{A}'$ by independently *dropping* existing edges with probability $p$ and *adding* non-edges with probability $q$:

$$A'_{ij} \sim \begin{cases} \mathrm{Bernoulli}(1-p), & A_{ij}=1 \\ \mathrm{Bernoulli}(q), & A_{ij}=0, \end{cases} \quad (6)$$

with $A'_{ji} = A'_{ij}$ and $A'_{ii} = 0$. Although we use Bernoulli perturbations, the framework can be extended to other distributions. RADE generalizes DropEdge: $q = 0$ recovers it, while $p = 0$ yields pure random edge addition.[4]

**Aggregation-correction mechanism.** A topology perturbation generally changes the expected aggregated messages, so maintaining train-inference alignment typically requires an explicit correction mechanism. RADE specifies at each layer how aggregation is defined on a perturbed view and how it is aligned with the inference-time aggregation. Under a weighted-sum aggregator, for a sampled view $\mathcal{G}'$, RADE aggregates via

$$\widetilde{\mathbf{a}}_i^{(\mathcal{G}',\ell)} = \sum_{j\neq i} \alpha_{ij}^{\mathcal{G}'} \, \widetilde{\mathbf{m}}_{ij}^{(\ell-1)}, \quad (7)$$

which corrects the training-time aggregation through the corrected message $\widetilde{\mathbf{m}}_{ij}^{(\ell-1)}$. Together with the inference-time aggregation to which it is aligned, this defines the alignment mechanism. We consider two RADE variants, depending on how the messages are corrected: RADE-OF, which targets only overfitting, and RADE-OFS, which targets both overfitting and over-squashing.

**RADE-OF.** When only overfitting is a concern, RADE-OF corrects messages on both existing edges and non-edges so

---

[2]For nonlinear GNNs, nonlinearity introduces bias after aggregation, so Eq. (5) holds in approximation.

[3]Ignoring cross-class covariances, Eq. (5) extends to $C$-class classification as $\mathbb{E}[\tilde{L}_{\mathrm{CE}}] \approx L_{\mathrm{CE}} + \frac{1}{2}\sum_i \sum_{c=1}^{C} p_{i,c}(1 - p_{i,c})\mathrm{Var}(\delta_{i,c})$, where $p_{i,c}$ is the predicted probability of class $c$ for node $i$, and $\delta_{i,c}$ is the class-$c$ logit perturbation.

[4]In practice, to avoid enumerating all non-edges $|\overline{E}|$, we add $K = q|\overline{E}|$ non-edges sampled uniformly without replacement–the hypergeometric analogue of i.i.d. Bernoulli($q$)–so $K/|\overline{E}| \approx q$.

that each layer's training-time aggregation is expectation-preserving with respect to the original inference-time aggregation on the input graph. Define

$$\widetilde{\mathbf{m}}_{ij}^{(\ell-1)} = \begin{cases} \dfrac{\alpha_{ij}^{\mathcal{G}}}{\mathbb{E}\left[\alpha_{ij}^{\mathcal{G}'}\right]} \, \mathbf{m}_{ij}^{(\ell-1)}, & A_{ij} = 1, \\ \mathbf{m}_{ij}^{(\ell-1)} - \boldsymbol{\mu}_i^{(\ell-1)}, & A_{ij} = 0, \end{cases} \quad (8)$$

where the centering term $\boldsymbol{\mu}_i^{(\ell-1)}$ is the weighted mean over non-neighbor messages:

$$\boldsymbol{\mu}_i^{(\ell-1)} = \frac{1}{Z_i} \sum_{j:A_{ij}=0} \mathbb{E}\left[\alpha_{ij}^{\mathcal{G}'}\right] \mathbf{m}_{ij}^{(\ell-1)}, \quad (9)$$

with $Z_i = \sum_{j:A_{ij}=0} \mathbb{E}\left[\alpha_{ij}^{\mathcal{G}'}\right]$. On existing edges ($A_{ij} = 1$), RADE-OF rescales messages so that each edge's expected contribution matches the coefficient $\alpha_{ij}^{\mathcal{G}}$. On non-edges ($A_{ij} = 0$), it centers added messages by $\boldsymbol{\mu}_i^{(\ell-1)}$, so that the total added contribution is zero in expectation.

**Proposition 4.1** (Expectation-Preservation in RADE-OF). *Under the RADE perturbation* (6)*, any weighted-sum aggregation (Def. 3.1) corrected by RADE-OF rule* (8) *is expectation-preserving (Def. 3.2).*

Thus, RADE-OF preserves each layer's message aggregation in expectation while still injecting stochasticity through the realized perturbations—the setting that underlies the variance-based regularization view, see Eq. (5).

**RADE-OFS.** When both overfitting and over-squashing are concerns, RADE-OFS uses stochastic edge deletion and addition during training, while modifying the inference-time aggregation to retain the effect of edge addition for mitigating over-squashing. Accordingly, its train-inference alignment target is a *modified inference-time aggregation* rather than the original input-graph aggregation.

During training, both random deletion and addition introduce stochasticity for regularization. The correction is designed so that, in expectation, the effect of random deletion is canceled, and deletion serves only as a source of stochastic regularization. The corrected messages are

$$\widetilde{\mathbf{m}}_{ij}^{(\ell-1)} = \begin{cases} \dfrac{\alpha_{ij}^{\mathcal{G}}}{\mathbb{E}\left[\alpha_{ij}^{\mathcal{G}'}\right]} \, \mathbf{m}_{ij}^{(\ell-1)}, & A_{ij} = 1 \\ \mathbf{m}_{ij}^{(\ell-1)}, & A_{ij} = 0 \end{cases} \quad (10)$$

Note that the effect of random edge addition is retained during training and later realized through inference-time aggregation, enabling the resulting densification to improve long-range communication. In inference, RADE-OFS aligns the corrected training-time aggregation, in expectation, with the modified aggregation of

$$\widehat{\mathbf{a}}_i^{(\mathcal{G},\ell)} = \mathbf{a}_i^{(\mathcal{G},\ell)} + \sum_{j:A_{ij}=0} \mathbb{E}\left[\alpha_{ij}^{\mathcal{G}'}\right] \mathbf{m}_{ij}^{(\ell-1)}, \quad (11)$$

*Table 1.* Comparison of graph augmentation settings for mitigating overfitting (OF), over-squashing (OS), or both (OFS).

| Goal | Graph Used | | Aggregation Correction | |
| | Training | Inference | Training | Inference |
| --- | --- | --- | --- | --- |
| OF | Stochastic $\mathcal{G}'$ | $\mathcal{G}$ | Yes | No |
| OS | Deterministic $\mathcal{G}'$ | $\mathcal{G}'$ | No | No |
| OFS | Stochastic $\mathcal{G}'$ | $\mathcal{G}$ | Yes | Yes |

which adds the deterministic expected contribution of non-edges and thereby creates additional communication paths that can mitigate over-squashing.

**Proposition 4.2** (Expectation-Preservation in RADE-OFS). *Under the RADE perturbation* (6)*, any weighted-sum aggregation (Def. 3.1) corrected by the RADE-OFS rule* (10) *is expectation-preserving with respect to the modified inference-time aggregation in Eq.* (11).

Thus, in RADE-OFS, the expected training-time aggregation matches the modified inference-time aggregation rather than the input-graph aggregation. Both random drop and add regularize training, while the drop effect is corrected in expectation, but the addition effect is retained at the inference aggregation rule to improve long-range communication.

Table 1 summarizes three augmentation settings and highlights where RADE-OF and RADE-OFS fit. RADE-OF corresponds to the augmentation settings for mitigating overfitting (OF), where stochastic perturbed views are used during training, inference uses the input graph, and training-time aggregation correction is required to remove train-inference mismatch. RADE-OFS corresponds to the setting for joint overfitting and over-squashing (OFS) mitigation, where training uses stochastic perturbed views, and inference remains on the input graph. However, both training-time and inference-time aggregations are corrected differently: the effect of edge removal (in expectation) is corrected away in training, while the expected effect of edge addition is kept in training and carried into inference through the modified inference-time aggregation. We note that rather than modifying/correcting the aggregation in inference, an alternative is to average predictions over multiple randomly perturbed graphs at inference. This would align training and inference through the same stochastic mechanism while defining an ensemble predictor over perturbed graphs. We avoid this approach for computational efficiency.

**Computing Expectation Terms in RADE.** For a weighted-sum backbone, RADE requires message-correction rules for aggregation alignment (see, e.g., Eqs. 8–11). These rules require expectation terms involving perturbed aggregation coefficients, such as $\mathbb{E}[\alpha_{ij}^{\mathcal{G}'}]$. When tractable, these terms can be derived analytically; when exact derivation is difficult, they can be approximated analytically or estimated

empirically by averaging realized coefficients over sampled perturbed views. Below, we provide analytic expressions or approximations for three common aggregators.

*Sum aggregation (GIN-style).* Here, $\alpha_{ij}^{\mathcal{G}} = A_{ij}$ and $\alpha_{ij}^{\mathcal{G}'} = A_{ij}'$, so $\mathbb{E}[\alpha_{ij}^{\mathcal{G}'}] = 1 - p$ on edges and $q$ on non-edges. Thus, in RADE-OF, the correction reduces to scalar rescaling on neighbors together with centering over non-neighbors, while in RADE-OFS, neighbors are rescaled in the same way, and non-neighbor contributions are retained and reintroduced at inference through Eq. (11).

*Symmetric normalization (GCN-style).* Here, $\alpha_{ij}^{\mathcal{G}} = A_{ij}(d_i d_j)^{-\frac{1}{2}}$ and $\alpha_{ij}^{\mathcal{G}'} = A_{ij}'(d_i' d_j')^{-\frac{1}{2}}$, where $d_i'$ and $d_j'$ are perturbed degrees. In this case, the required expectation terms depend on random degrees and are harder to derive exactly. Moreover, they vary across node pairs and therefore do not reduce to a global scaling factor. In Appendix C, we derive analytic approximations via mixed-binomial delta-method expansions. More generally, these terms can also be estimated empirically from sampled perturbed views.

*Attention normalization (GAT-style).* For attention aggregation, $\alpha_{ij}^{\mathcal{G}} = \frac{A_{ij} \exp(\beta_{ij})}{\sum_{k \neq i} A_{ik} \exp(\beta_{ik})}$, where $\beta_{ij}$ is raw attention score of $i$ to $j$. Conditional on the previous-layer representations, the raw attention scores are fixed, but the graph perturbation changes both the numerator and the denominator. Also, $\mathbb{E}[\alpha_{ij}^{\mathcal{G}'}]$ is pair-dependent. In Appendix C, we present an approximation method for expectation terms combining the delta-method expansion and sampling method.

**Logit Variance.** As seen in Eq. (5), the regularization effect of stochastic graph perturbation is controlled by the logit variance $\mathrm{Var}(\delta_i)$. Thus, to understand how RADE regularizes, we analyze $\mathrm{Var}(\delta_i)$ under RADE's perturbation.

**Proposition 4.3** (Logit perturbation variance). *Assuming for each node $i$ the stochastic aggregation coefficients $\alpha_{ij}^{\mathcal{G}'}$ are mutually independent across $j$,[5] then*

$$\mathrm{Var}(\delta_i) = \sum_{j \neq i} \widetilde{m}_{ij}^2 \, \mathrm{Var}\left(\alpha_{ij}^{\mathcal{G}'}\right). \tag{12}$$

*Multiclass results hold coordinate-wise.*

This shows that the variability in aggregation coefficients $\alpha_{ij}^{\mathcal{G}'}$—induced by graph perturbation—determines logit variance $\mathrm{Var}(\delta_i)$, which induces the implicit regularization.[6]

From Proposition 4.3, one can decompose $\mathrm{Var}(\delta_i)$ into two additive terms with disjoint support: one over neighbors

and one over non-neighbors. These two terms isolate how deletion and addition contribute to the regularizer, through variability induced by edges and non-edges. This separability of addition and drop terms also lets us study the relationship between deletion and addition: whether or not they induce interchangeable regularization effects.

**Add-Drop Non-interchangeability.** We examine whether each primitive (e.g., Add-only or Drop-only) can subsume the other to understand their interchangeability.

**Proposition 4.4** (Restricted interchangeability of Drop-only and Add-only). *For sum aggregation[7], Drop-only and Add-only only subsume each other by matching their node-wise variances under a restrictive setting in which there should exist a constant $\kappa > 0$ such that $\sum_{j:A_{ij}=0} \widetilde{m}_{ij}^2 = \kappa \sum_{j:A_{ij}=1} m_{ij}^2$ for every node $i$.*

The proposition identifies major barriers to exact interchangeability. Drop-only and Add-only act on different node-wise statistics. Drop-only scales the existing-edge statistic $\sum_{j:A_{ij}=1} m_{ij}^2$, whereas Add-only scales a non-edge statistic: $\sum_{j:A_{ij}=0}(m_{ij} - \mu_i)^2$ for RADE-OF and $\sum_{j:A_{ij}=0} m_{ij}^2$ for RADE-OFS. Thus, a single global addition rate can match a Drop-only perturbation, node by node, for all nodes only when these two statistics are proportional with the same constant for every node in the graph.[8] Our ablation studies confirm that add and drop are not only non-interchangeable, but also complementary in practice.

**Adaptive Control of Regularization Strength.** We have shown that RADE regularizes through the variance of the logit perturbation by perturbing the graph. The remaining question is *how strongly* RADE should regularize during training, which is controlled by perturbation hyperparameters $p$ and $q$ for deletions and additions. This is a practical issue in stochastic graph augmentation: if the perturbation is too weak, its regularization effect is negligible; if it is too strong, the sampled views may deviate excessively from the input graph and distort the message-passing structure. Existing augmentation methods typically handle this trade-off through dataset-specific tuning of perturbation hyperparameters. In contrast, we propose a principled rule that adapts the deletion and addition rates during training and reduces reliance on costly rate sweeps.

We draw inspiration from GradNorm (Chen et al., 2018c), which balances multiple objectives by matching gradient norms on shared parameters. We treat RADE's variance-

---

[5]This independence assumption holds when the randomness entering the coefficients is separate across $j$ (e.g., sum aggregation), and is an approximation for degree-normalized coefficients where they share randomness through perturbed degrees.

[6]Proposition 4.3 extends to graph classification, assuming linear graph pooling and prediction head.

[7]We focus on sum aggregation because the variance separates into a rate-dependent scalar and a node-wise message statistic, yielding a simple condition for comparing Drop- and Add-only.

[8]Our theory establishes the limited statement that Drop-only and Add-only are generally non-interchangeable. It does not characterize the full set of variance profiles attainable by joint add-drop perturbations, which we leave to future work.

based regularization effect as an (implicit) objective and match its gradient scale to that of the supervised loss by adjusting p and q. For a mini-batch $B$, define

$$R(B, p, q) = \frac{1}{2} \sum_{i \in B} z_i (1 - z_i) \operatorname{Var}(\delta_i), \qquad (13)$$

as the RADE regularization proxy. We measure the gradient magnitudes of the supervised loss $L_{\text{data}}$ and the RADE regularization proxy on shared parameters $\boldsymbol{\theta}$: $G_{\text{data}}^B = \|\nabla_{\boldsymbol{\theta}} L_{\text{data}}(B)\|_2$, and $G_{\text{reg}}^B(p, q) = \|\nabla_{\boldsymbol{\theta}} R(B, p, q)\|_2$. The deletion and addition rates are then updated by minimizing

$$\mathcal{J}(p, q) = \left[ \log \left( \frac{G_{\text{reg}}^B(p, q) + \epsilon}{G_{\text{data}}^B + \epsilon} \right) \right]^2 + \lambda \left( \frac{q}{D(\mathcal{G})} \right)^2. \quad (14)$$

The first term is the gradient-matching objective: it is minimized when the RADE regularization and the supervised loss induce comparable gradient magnitudes on the shared parameters. The logarithmic ratio matches relative scale rather than absolute difference, and $\epsilon > 0$ stabilizes the objective when either gradient norm is zero. The second term penalizes excessive edge addition. Since $q$ is an addition probability over many non-edges, even a small $q$ can add many edges, thus changing the sparse regime of the input graph to dense. Normalizing by the graph density[9] $D(\mathcal{G})$ makes this penalty depend on the addition rate relative to the original graph density, thus discouraging densification and preserving the inductive bias of the original input graph. The coefficient $\lambda$ controls the density-normalized penalty of the second term: larger values make the model more conservative in adding edges. To avoid introducing dataset-specific hyperparameters, we fix $\lambda$ at the task level (e.g., graph or node classification) rather than tuning it per dataset; dataset-specific adaptation is through the GradNorm rate updates.[10]

In practice, we take one optimization step on $\mathcal{J}(p, q)$ after each mini-batch, and use the updated rates $(p, q)$ to sample the perturbation for the next mini-batch. This makes RADE *adaptive* throughout training: the perturbation rates evolve during the optimization rather than being fixed hyperparameters. The update increases perturbation strength when the regularization gradient is too weak, and decreases it when the regularization gradient becomes dominant. This directly addresses a key limitation of stochastic graph augmentation: its sensitivity to manually-chosen fixed perturbation rates. Further implementation details are in Appendix G.

---

[9]Graph density is the fraction of the number of edges to all possible edges, measuring the extent of sparsity.

[10]For optimizing $\mathcal{J}$, we reparameterize the addition rate $q$ by setting $\rho = q/D(\mathcal{G})$ and optimize $\rho$ instead, mapping back via $q = \rho D(\mathcal{G})$ when evaluating $G_{\text{reg}}^B(p, q)$ or sampling edges. This doesn't change the objective or perturbation rule; it only rescales the optimization coordinate through a variable change (Boyd & Vandenberghe, 2004). Since even a small change in the raw non-

## 5. Experiments

Our experiments evaluate RADE-OF and RADE-OFS on various node classification and graph classification tasks.[11]

**Datasets.** For node classification, we evaluate on eight datasets: Cora, CiteSeer, and PubMed (Sen et al., 2008); Computer, CS, and Physics (Shchur et al., 2018); Flickr (Zeng et al., 2020); and ogbn-arxiv (Hu et al., 2020). We use public splits for Cora, CiteSeer, PubMed, Flickr, and ogbn-arxiv; random 60%/20%/20% train/validation/test splits for Computer, CS, and Physics, following (Luo et al., 2024). For graph classification, we evaluate on MUTAG, PRO-TEINS, IMDB-B, and IMDB-M from TU (Morris et al., 2020), ogbg-molhiv from OGB (Hu et al., 2020), and peptides-func from LRGB (Dwivedi et al., 2022). For TU datasets, we follow a 10-fold cross-validation protocol (Xu et al., 2019); ogbg-molhiv and peptides-func use their official benchmark splits. Dataset statistics are in Appendix I.

**Baselines.** We compare RADE to widely-used regularization and augmentation baselines: Dropout (Srivastava et al., 2014), DropEdge (Rong et al., 2020), DropNode (Feng et al., 2020), and DropMessage (Fang et al., 2023). All methods use the same backbone and training pipeline: GCN, GIN, or GAT. Some results are deferred to Appendix H.

**Experimental Setup.** For node classification, we follow recent protocols (Luo et al., 2024; Lee et al., 2025), with minor modifications to isolate the effect of graph augmentation by disabling other auxiliary regularizers (e.g., weight decay, dropout, batch normalization). We use 2-layer backbones, a common setup for node classification (Ju et al., 2023; Fang et al., 2023; Luo et al., 2024). For each dataset, we tune the backbone hidden dimension without augmentation and fix it across all regularizers. For augmentation baselines, we tune their corresponding augmentation rate (e.g., drop rate for DropEdge). We train with Adam using a learning rate 0.001, using full-batch training except for Flickr and ogbn-arxiv with mini-batch training. We apply early stopping with a patience of 100. We report mean±std over five runs, using accuracy for all node-classification datasets except Flickr, where we report micro-F1. For RADE, we always initialize $p = 0.5$ and $q$ with the graph density; during training, $p$ and $q$ are adapted by one Adam step on the GradNorm objective $\mathcal{J}$ with learning rate 0.001. For graph classification, we follow dataset-specific standard protocols. On TU datasets, we follow the GIN protocol (Xu et al., 2019): 4 message-passing layers and 10-fold cross-validation. For ogbg-molhiv, we follow the OGB protocol and hyperparameters (Hu et al., 2020); for peptides-func,

---

edge probability $q$ can induce substantial densification, optimizing $\rho$ instead makes updates relative to the graph density.

[11]Code is available at https://github.com/Danial-sb/RADE

*Table 2.* Node classification results (mean±std over five runs) with GCN backbone. Accuracy (%) on all datasets except Flickr with micro-F1 (%). Best in bold and second underlined. Green and red shading show improvement and degradation over the GCN baseline.

| Method | Cora | CiteSeer | PubMed | CS | Computer | Physics | Flickr | ogbn-arxiv |
|---|---|---|---|---|---|---|---|---|
| GCN | $80.10 \pm 0.64$ | $69.12 \pm 1.10$ | $77.49 \pm 0.42$ | $92.97 \pm 0.18$ | $89.63 \pm 0.24$ | $96.52 \pm 0.05$ | $51.89 \pm 0.16$ | $70.51 \pm 0.14$ |
| Dropout | $80.76 \pm 0.85$ | $70.04 \pm 0.58$ | $77.70 \pm 0.72$ | $92.92 \pm 0.10$ | $89.71 \pm 0.27$ | $\underline{96.54 \pm 0.04}$ | $52.02 \pm 0.10$ | $70.77 \pm 0.42$ |
| DropEdge | $80.28 \pm 0.34$ | $69.42 \pm 1.16$ | $77.80 \pm 0.38$ | $92.10 \pm 0.27$ | $89.79 \pm 0.31$ | $96.48 \pm 0.06$ | $52.08 \pm 0.10$ | $70.38 \pm 0.27$ |
| DropNode | $80.80 \pm 0.55$ | $68.42 \pm 1.43$ | $77.68 \pm 0.64$ | $92.96 \pm 0.14$ | $89.63 \pm 0.22$ | $96.46 \pm 0.06$ | $52.07 \pm 0.12$ | $70.55 \pm 0.32$ |
| DropMessage | $80.65 \pm 1.01$ | $67.94 \pm 1.82$ | $77.57 \pm 0.40$ | $91.57 \pm 0.33$ | $89.06 \pm 0.64$ | $96.50 \pm 0.05$ | $\underline{52.23 \pm 0.13}$ | $70.45 \pm 0.29$ |
| RADE-OF | $\mathbf{81.08 \pm 1.06}$ | $\mathbf{70.20 \pm 1.31}$ | $\underline{78.04 \pm 0.80}$ | $\mathbf{93.26 \pm 0.72}$ | $\mathbf{90.22 \pm 0.14}$ | $\mathbf{96.58 \pm 0.10}$ | $\mathbf{52.25 \pm 0.25}$ | $\mathbf{71.09 \pm 0.26}$ |
| RADE-OFS | $\underline{80.82 \pm 1.24}$ | $\underline{70.12 \pm 2.25}$ | $\mathbf{78.24 \pm 0.85}$ | $\underline{93.21 \pm 0.15}$ | $89.94 \pm 0.50$ | $96.53 \pm 0.03$ | $51.92 \pm 0.12$ | $\underline{70.95 \pm 0.19}$ |

*Table 3.* Graph classification results (mean±std) with GCN backbone. Accuracy (%) on TU, ROC-AUC (%) on ogbg-molhiv, and AP (%) on peptides-func. Best in bold and second underlined. Green/red shading shows improvements/degradations vs. GCN baseline.

| Method | MUTAG | PROTEINS | IMDB-B | IMDB-M | ogbg-molhiv | peptides-func |
|---|---|---|---|---|---|---|
| GCN | $84.00 \pm 8.00$ | $75.37 \pm 0.15$ | $75.70 \pm 3.13$ | $51.93 \pm 2.15$ | $75.21 \pm 1.24$ | $55.14 \pm 0.54$ |
| Dropout | $83.97 \pm 4.98$ | $75.40 \pm 4.41$ | $75.80 \pm 2.40$ | $52.06 \pm 2.11$ | $74.00 \pm 0.94$ | $54.84 \pm 0.45$ |
| DropEdge | $84.50 \pm 7.09$ | $75.29 \pm 2.74$ | $75.70 \pm 3.25$ | $51.60 \pm 2.84$ | $76.01 \pm 0.54$ | $55.29 \pm 0.88$ |
| DropNode | $85.02 \pm 5.99$ | $75.01 \pm 4.29$ | $76.20 \pm 4.12$ | $\underline{52.20 \pm 3.04}$ | $76.04 \pm 1.00$ | $54.21 \pm 0.76$ |
| DropMessage | $86.08 \pm 7.39$ | $75.45 \pm 3.80$ | $75.40 \pm 3.41$ | $51.33 \pm 2.92$ | $74.21 \pm 1.07$ | $54.98 \pm 0.60$ |
| RADE-OF | $\underline{86.16 \pm 5.62}$ | $\underline{75.62 \pm 1.86}$ | $\underline{76.20 \pm 3.24}$ | $52.09 \pm 2.21$ | $\underline{76.09 \pm 1.24}$ | $\underline{56.12 \pm 0.66}$ |
| RADE-OFS | $\mathbf{86.24 \pm 6.79}$ | $\mathbf{75.84 \pm 3.44}$ | $\mathbf{76.60 \pm 2.25}$ | $\mathbf{52.24 \pm 1.95}$ | $\mathbf{76.28 \pm 1.41}$ | $\mathbf{56.97 \pm 0.82}$ |

we follow the LRGB setup (Dwivedi et al., 2022). We report accuracy for TU, ROC-AUC for ogbg-molhiv, and AP for peptides-func. Dataset-specific hyperparameters are provided in Appendix I.

**Results.** On node classification (Table 2), both RADE variants improve upon the GCN baseline on all datasets. RADE-OF is the strongest overall variant: it is best on seven datasets and second-best on PubMed. Compared with the strongest non-RADE baseline, its gains range from 0.02% (Flickr) to 0.43% (Computer). RADE-OFS is best on PubMed and is otherwise below RADE-OF, consistent with its design: retaining the non-edge contribution at inference can help when extra test-time propagation mitigates over-squashing.[12] Overall, since these node-classification datasets are less affected by over-squashing (Saber & Salehi-Abari, 2025), RADE-OF is the more reliable default.

On graph classification (Table 3), RADE-OFS is best on all datasets and improves over vanilla GCN on every dataset, with the largest gains on MUTAG (+2.24%) and peptides-func (+1.83%). RADE-OF also improves over GCN on all datasets; it is second-best on MUTAG, PROTEINS, ogbg-molhiv, and peptides-func, ties for second on IMDB-B, and is below DropNode on IMDB-M. These results show that both RADE variants are reliable regularizers, and also support the role of RADE-OFS in graph-level tasks, where over-squashing is more pronounced (Saber & Salehi-Abari,

2025). By retaining expected non-edge contributions at inference, RADE-OFS creates additional propagation paths, aligning with its strongest gains on datasets that require long-range interaction or exhibit over-squashing (Dwivedi et al., 2022; Saber & Salehi-Abari, 2025).

**Ablations: GradNorm, Corrections, and Nonlinearity.** Table 4 studies three components of RADE-OF: adaptive rate selection through GradNorm (GN), expectation-preserving correction (EPC), and nonlinear message passing. Removing GradNorm (w/o GN), which fixes the drop/add rates to their default initialization, reduces performance on most datasets, with the largest drop of $-0.83\%$ on Flickr. In the fixed-rate setting, removing both GradNorm and the expectation-preserving correction (w/o GN & EPC) typically degrades performance more substantially (e.g., $-2.90\%$ on Flickr), highlighting the importance of aggregation corrections under stochastic drop/add perturbations.

Table 4 also reports RADE-OF-Lin, with a linearized GCN backbone—known as SGC (Wu et al., 2019a)—matching the linear setting in Eq. (5). RADE-OF-Lin improves SGC on all datasets, showing that RADE remains effective in the linearized setting. However, nonlinear RADE-OF is consistently stronger, especially on Flickr and ogbn-arxiv. Removing GN and EPC from RADE-OF-Lin also weakens performance on most datasets. Overall, the ablations confirm the contributing value of adaptive rate selection, expectation-preserving corrections, and nonlinearity (despite violating the linearity assumption of Eq. 5).

---

[12]For PubMed, this aligns with its reported moderate over-squashing intensity (Saber & Salehi-Abari, 2025).

*Table 4.* Node-classification ablations of GradNorm (GN), Expectation-Preserving Correction (EPC), and nonlinearity for RADE-OF (mean±std over five runs) with GCN backbone. Accuracy (%) is reported for all datasets except Flickr with micro-F1 (%). RADE-OF-Lin and GCN-Lin have the linearized GCN backbone. Best is in bold.

| Method | Cora | CiteSeer | PubMed | CS | Computer | Physics | Flickr | ogbn-arxiv |
|---|---|---|---|---|---|---|---|---|
| RADE-OF | $\mathbf{81.08_{\pm1.06}}$ | $\mathbf{70.20_{\pm1.31}}$ | $\mathbf{78.04_{\pm0.80}}$ | $\mathbf{93.26_{\pm0.72}}$ | $\mathbf{90.22_{\pm0.14}}$ | $\mathbf{96.58_{\pm0.10}}$ | $\mathbf{52.25_{\pm0.25}}$ | $\mathbf{71.09_{\pm0.26}}$ |
| RADE-OF w/o GN | $80.90_{\pm1.32}$ | $70.05_{\pm0.45}$ | $77.43_{\pm0.88}$ | $93.10_{\pm0.22}$ | $90.15_{\pm0.32}$ | $96.53_{\pm0.12}$ | $51.42_{\pm0.20}$ | $70.52_{\pm0.35}$ |
| RADE-OF w/o GN & EPC | $78.94_{\pm0.50}$ | $70.16_{\pm1.10}$ | $77.18_{\pm0.76}$ | $93.09_{\pm0.25}$ | $89.04_{\pm0.42}$ | $96.52_{\pm0.12}$ | $49.35_{\pm0.12}$ | $70.33_{\pm0.32}$ |
| RADE-OF-Lin | $80.42_{\pm0.54}$ | $68.76_{\pm1.05}$ | $76.96_{\pm0.52}$ | $93.22_{\pm0.13}$ | $89.32_{\pm0.30}$ | $96.47_{\pm0.05}$ | $50.65_{\pm0.18}$ | $69.92_{\pm0.35}$ |
| RADE-OF-Lin w/o GN | $80.06_{\pm0.75}$ | $68.44_{\pm0.72}$ | $76.84_{\pm1.25}$ | $93.07_{\pm0.21}$ | $88.32_{\pm0.21}$ | $96.45_{\pm0.04}$ | $50.12_{\pm0.25}$ | $69.05_{\pm0.12}$ |
| RADE-OF-Lin w/o GN & EPC | $77.80_{\pm0.50}$ | $68.91_{\pm0.66}$ | $76.72_{\pm1.01}$ | $92.72_{\pm0.26}$ | $88.25_{\pm0.25}$ | $96.45_{\pm0.03}$ | $50.09_{\pm0.29}$ | $68.72_{\pm0.18}$ |
| GCN-Lin (SGC) | $79.60_{\pm0.25}$ | $68.62_{\pm0.41}$ | $75.24_{\pm0.65}$ | $93.01_{\pm0.19}$ | $88.28_{\pm0.31}$ | $96.44_{\pm0.08}$ | $50.12_{\pm0.26}$ | $68.67_{\pm0.32}$ |

*Table 5.* Drop-only vs. add-only decomposition of RADE-OF on node classification (mean±std over five runs) with GCN backbone. Accuracy (%) is reported on all datasets except Flickr with micro-F1 (%). Best is in bold.

| Method | Cora | CiteSeer | PubMed | CS | Computer | Physics | Flickr | ogbn-arxiv |
|---|---|---|---|---|---|---|---|---|
| RADE-OF | $\mathbf{81.08 \pm 1.06}$ | $\mathbf{70.20 \pm 1.31}$ | $\mathbf{78.04 \pm 0.80}$ | $\mathbf{93.26 \pm 0.72}$ | $\mathbf{90.22 \pm 0.14}$ | $\mathbf{96.58 \pm 0.10}$ | $\mathbf{52.25 \pm 0.25}$ | $\mathbf{71.09 \pm 0.26}$ |
| RADE-OF-Drop | $80.44 \pm 1.26$ | $69.84 \pm 0.82$ | $77.82 \pm 1.27$ | $92.16 \pm 0.25$ | $89.82 \pm 0.34$ | $96.48 \pm 0.20$ | $52.14 \pm 0.15$ | $70.68 \pm 0.15$ |
| RADE-OF-Add | $80.68 \pm 0.75$ | $69.34 \pm 1.65$ | $77.96 \pm 0.52$ | $92.94 \pm 0.24$ | $90.09 \pm 0.31$ | $96.49 \pm 0.06$ | $51.85 \pm 0.15$ | $70.12 \pm 0.18$ |

*Table 6.* Fixed fine-tuned vs. adaptive perturbation rates on node classification (mean±std over five runs) with GCN backbone. "Tuned" selects a single best rate pair by hyperparameter search, while RADE-OF/RADE-OFS use adaptive GradNorm rule. Best is in bold.

| Method | Cora | CiteSeer | PubMed | CS | Computer | Physics | Flickr | ogbn-arxiv |
|---|---|---|---|---|---|---|---|---|
| RADE-OF | $\mathbf{81.08 \pm 1.06}$ | $\mathbf{70.20 \pm 1.31}$ | $\mathbf{78.04 \pm 0.80}$ | $\mathbf{93.26 \pm 0.72}$ | $\mathbf{90.22 \pm 0.14}$ | $\mathbf{96.58 \pm 0.10}$ | $\mathbf{52.25 \pm 0.25}$ | $\mathbf{71.09 \pm 0.26}$ |
| RADE-OF (tuned) | $80.90 \pm 1.32$ | $70.08 \pm 0.62$ | $77.43 \pm 0.88$ | $93.12 \pm 0.06$ | $90.15 \pm 0.32$ | $96.53 \pm 0.08$ | $52.15 \pm 0.18$ | $70.56 \pm 0.36$ |
| RADE-OFS | $\mathbf{80.82 \pm 1.24}$ | $\mathbf{70.12 \pm 2.25}$ | $\mathbf{78.24 \pm 0.85}$ | $\mathbf{93.21 \pm 0.15}$ | $\mathbf{89.94 \pm 0.50}$ | $\mathbf{96.53 \pm 0.03}$ | $\mathbf{51.92 \pm 0.12}$ | $\mathbf{70.95 \pm 0.19}$ |
| RADE-OFS (tuned) | $80.20 \pm 0.84$ | $69.74 \pm 1.43$ | $77.82 \pm 0.48$ | $92.20 \pm 0.10$ | $88.31 \pm 0.53$ | $96.49 \pm 0.06$ | $51.84 \pm 0.18$ | $70.77 \pm 0.25$ |

**Decomposing RADE: Drop and Add.** Table 5 decomposes RADE-OF into its drop and add components. The combined add-drop variant outperforms both single-component variants on all datasets. The relative strength of drop-only and add-only differs across datasets—add-only is stronger on Cora, PubMed, CS, Computer, and Physics, while drop-only is stronger on CiteSeer, Flickr, and ogbn-arxiv. This supports that deletion and addition are not interchangeable, and combining them yields better results. RADE-OF-Drop also improves over DropEdge in Table 2, since it retains adaptive rate selection and expectation-preserving correction.

**Adaptive vs. Fine-tuned Rates.** Table 6 compares adaptive selection of $(p, q)$ with a fine-tuned fixed pair. GradNorm improves over fixed fine-tuned rates on all datasets for both RADE variants, with modest but consistent gains for RADE-OF and larger gains for RADE-OFS. This supports adapting perturbation strength during training rather than using hyperparameter-tuned rates.

## 6. Conclusion

We presented RADE, a stochastic add-drop edge augmentation that jointly addresses overfitting and over-squashing.

We introduced two variants. RADE-OF targets overfitting by aligning the expected training-time aggregation with the input-graph inference aggregation, so random perturbations act only as regularization. RADE-OFS targets both overfitting and over-squashing by correcting deletion effects while retaining expected added-edge contributions at inference. We showed RADE's deletion and addition components induce non-interchangeable regularization effects, and introduced a mini-batch GradNorm rule that adapts their rates during training. Across node and graph classification tasks, RADE-OF is a reliable regularizer, while RADE-OFS yields the clearest gains when long-range communication matters. Ablations support the importance of expectation-preserving corrections, adaptive rate selection, and combining deletion with addition.

## Acknowledgements

This work was supported by the Natural Sciences and Engineering Research Council of Canada (NSERC). We also thank the reviewers for their constructive comments and suggestions.

## Impact Statement

This paper proposes a graph data augmentation and regularization method intended to improve the training of graph neural networks. The method is domain-agnostic and does not target any specific application area. We do not anticipate unique negative societal or ethical impacts arising specifically from this method.

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

## A. Derivation of the Variance-Regularization View

For completeness, we briefly derive the approximation in Eq. (5), following the same second-order argument used in prior work on stochastic regularization for GNNs (Fang et al., 2023).

**Binary cross-entropy.** Let $s_i$ denote the logit of node $i$ on the input graph, and let $\tilde{s}_i$ denote the corresponding logit on the perturbed graph. We observe $\tilde{s}_i = s_i + \delta_i$ with logit noise of $\delta_i$. We assume that the graph augmentation/perturbation is expectation-preserving at the logit level, so that $\mathbb{E}[\delta_i] = 0$. We denote the binary cross-entropy (BCE) loss of the input graph by $L_{\mathrm{BCE}}$, while letting $\tilde{L}_{\mathrm{BCE}}$ denote the binary cross-entropy loss evaluated on the perturbed graph. We write $L_{\mathrm{BCE}} = \sum_i \ell_i(s_i)$ and $\tilde{L}_{\mathrm{BCE}} = \sum_i \ell_i(\tilde{s}_i)$ by defining

$$\ell_i(s_i) = -y_i \log z_i - (1 - y_i) \log(1 - z_i), \qquad z_i = \sigma(s_i),$$

Using a second-order Taylor expansion of $\ell_i(s_i + \delta_i)$ around $s_i$, we have $\ell_i(s_i + \delta_i) \approx \ell_i(s_i) + \ell_i'(s_i)\delta_i + \frac{1}{2}\ell_i''(s_i)\delta_i^2$. Taking the expectation over the augmentation randomness gives

$$\mathbb{E}[\ell_i(s_i + \delta_i)] \approx \ell_i(s_i) + \ell_i'(s_i)\mathbb{E}[\delta_i] + \frac{1}{2}\ell_i''(s_i)\mathbb{E}[\delta_i^2].$$

Since $\mathbb{E}[\delta_i] = 0$, $\mathbb{E}[\delta_i^2] = \mathrm{Var}(\delta_i)$, and $\ell_i''(s_i) = z_i(1 - z_i)$ for BCE, we have $\mathbb{E}[\ell_i(\tilde{s}_i)] \approx \ell_i(s_i) + \frac{1}{2}z_i(1 - z_i)\mathrm{Var}(\delta_i)$. Summing over $i$ yields

$$\mathbb{E}[\tilde{L}_{\mathrm{BCE}}] \approx L_{\mathrm{BCE}} + \frac{1}{2}\sum_i z_i(1 - z_i)\mathrm{Var}(\delta_i),$$

which is Eq. (5). This shows that stochastic expectation-preserving augmentation acts as a variance-based regularizer.

**Multi-class cross-entropy.** The same argument extends to $C$-class classification. Let $\mathbf{s}_i \in \mathbb{R}^C$ be the logits of node $i$ on the input graph, and let $\tilde{\mathbf{s}}_i$ denote the corresponding logits on the perturbed graph. We write $\tilde{\mathbf{s}}_i = \mathbf{s}_i + \boldsymbol{\delta}_i$, with the assumption of expectation-preservation at the logit level, so that $\mathbb{E}[\boldsymbol{\delta}_i] = \mathbf{0}$. Letting $\mathbf{p}_i = \mathrm{softmax}(\mathbf{s}_i)$, we consider the cross-entropy (CE) losses of $L_{\mathrm{CE}} = \sum_i \ell_i(\mathbf{s}_i)$ and $\widetilde{L}_{\mathrm{CE}} = \sum_i \ell_i(\tilde{\mathbf{s}}_i)$, with $\ell_i(\mathbf{s}_i) = -\log p_{i,y_i}$. Using a second-order Taylor expansion of $\ell_i(\mathbf{s}_i + \boldsymbol{\delta}_i)$ around $\mathbf{s}_i$, we have

$$\ell_i(\mathbf{s}_i + \boldsymbol{\delta}_i) \approx \ell_i(\mathbf{s}_i) + \nabla\ell_i(\mathbf{s}_i)^\top \boldsymbol{\delta}_i + \frac{1}{2}\boldsymbol{\delta}_i^\top \mathbf{H}_i \boldsymbol{\delta}_i,$$

where $\mathbf{H}_i = \mathrm{diag}(\mathbf{p}_i) - \mathbf{p}_i \mathbf{p}_i^\top$. Taking expectation and using $\mathbb{E}[\boldsymbol{\delta}_i] = \mathbf{0}$, we have $\mathbb{E}[\ell_i(\tilde{\mathbf{s}}_i)] \approx \ell_i(\mathbf{s}_i) + \frac{1}{2}\mathbb{E}\left[\boldsymbol{\delta}_i^\top \mathbf{H}_i \boldsymbol{\delta}_i\right]$. As the perturbation is mean-zero, we have $\mathbb{E}\left[\boldsymbol{\delta}_i^\top \mathbf{H}_i \boldsymbol{\delta}_i\right] = \mathrm{Tr}(\mathbf{H}_i \boldsymbol{\Sigma}_i)$, where $\boldsymbol{\Sigma}_i = \mathrm{Cov}(\boldsymbol{\delta}_i)$ is the covariance matrix. Thus,

$$\mathbb{E}[\tilde{L}_{\mathrm{CE}}] \approx L_{\mathrm{CE}} + \frac{1}{2}\sum_i \mathrm{Tr}(\mathbf{H}_i \boldsymbol{\Sigma}_i).$$

Ignoring cross-class covariances, we approximate $\boldsymbol{\Sigma}_i = \mathrm{diag}(\mathrm{Var}(\delta_{i,1}), \ldots, \mathrm{Var}(\delta_{i,C}))$, then $\mathrm{Tr}(\mathbf{H}_i \boldsymbol{\Sigma}_i) = \sum_{c=1}^{C} p_{i,c}(1 - p_{i,c})\mathrm{Var}(\delta_{i,c})$, and thus

$$\mathbb{E}[\tilde{L}_{\mathrm{CE}}] \approx L_{\mathrm{CE}} + \frac{1}{2}\sum_i \sum_{c=1}^{C} p_{i,c}(1 - p_{i,c})\mathrm{Var}(\delta_{i,c}).$$

## B. Proofs of Expectation-Preserving Aggregation of RADE-OF and RADE-OFS

Throughout this section, expectations are taken over $\mathcal{G}' \sim \mathcal{Q}(\cdot \mid \mathcal{G})$ conditional on $\mathbf{H}^{(\ell-1)}$. Thus, the messages $\{\mathbf{m}_{ij}^{(\ell-1)}\}_{j \neq i}$ are deterministic under the expectation. For notational brevity, we omit this conditioning in the proofs below. We first provide two lemmas and their proofs, which are used to prove our results.

**Lemma B.1** (Existing-edge correction). *Fix a node $i$ and layer $\ell$. Consider that each message from the existing edge $(i, j)$ is corrected by $\widetilde{\mathbf{m}}_{ij}^{(\ell-1)} = \frac{\alpha_{ij}^{\mathcal{G}}}{\mathbb{E}[\alpha_{ij}^{\mathcal{G}'}]}\mathbf{m}_{ij}^{(\ell-1)}$, with $\mathbb{E}[\alpha_{ij}^{\mathcal{G}'}] \neq 0$. Then $\mathbb{E}\left[\sum_{j:A_{ij}=1} \alpha_{ij}^{\mathcal{G}'}\widetilde{\mathbf{m}}_{ij}^{(\ell-1)}\right] = \sum_{j:A_{ij}=1} \alpha_{ij}^{\mathcal{G}}\mathbf{m}_{ij}^{(\ell-1)}$.*

*Proof.* Using linearity of expectation and the definition of $\widetilde{\mathbf{m}}_{ij}^{(\ell-1)}$,

$$\mathbb{E}\left[\sum_{j:A_{ij}=1}\alpha_{ij}^{\mathcal{G}'}\widetilde{\mathbf{m}}_{ij}^{(\ell-1)}\right] = \sum_{j:A_{ij}=1}\mathbb{E}\left[\alpha_{ij}^{\mathcal{G}'}\frac{\alpha_{ij}^{\mathcal{G}}}{\mathbb{E}[\alpha_{ij}^{\mathcal{G}'}]}\mathbf{m}_{ij}^{(\ell-1)}\right] = \sum_{j:A_{ij}=1}\mathbb{E}[\alpha_{ij}^{\mathcal{G}'}]\frac{\alpha_{ij}^{\mathcal{G}}}{\mathbb{E}[\alpha_{ij}^{\mathcal{G}'}]}\mathbf{m}_{ij}^{(\ell-1)} = \sum_{j:A_{ij}=1}\alpha_{ij}^{\mathcal{G}}\mathbf{m}_{ij}^{(\ell-1)}.$$

$\square$

**Lemma B.2** (Weighted centering identity). *Fix a node $i$ and layer $\ell$. Let $Z_i = \sum_{j:A_{ij}=0}\mathbb{E}[\alpha_{ij}^{\mathcal{G}'}]$, and, when $Z_i > 0$, let*
$\boldsymbol{\mu}_i^{(\ell-1)} = \frac{1}{Z_i}\sum_{j:A_{ij}=0}\mathbb{E}[\alpha_{ij}^{\mathcal{G}'}]\mathbf{m}_{ij}^{(\ell-1)}$. *Then $\sum_{j:A_{ij}=0}\mathbb{E}[\alpha_{ij}^{\mathcal{G}'}]\left(\mathbf{m}_{ij}^{(\ell-1)} - \boldsymbol{\mu}_i^{(\ell-1)}\right) = \mathbf{0}$.*

*Proof.* Substituting the definition of $\boldsymbol{\mu}_i^{(\ell-1)}$,

$$\begin{aligned}
\sum_{j:A_{ij}=0}\mathbb{E}[\alpha_{ij}^{\mathcal{G}'}]\left(\mathbf{m}_{ij}^{(\ell-1)} - \boldsymbol{\mu}_i^{(\ell-1)}\right) &= \sum_{j:A_{ij}=0}\mathbb{E}[\alpha_{ij}^{\mathcal{G}'}]\mathbf{m}_{ij}^{(\ell-1)} - \left(\sum_{j:A_{ij}=0}\mathbb{E}[\alpha_{ij}^{\mathcal{G}'}]\right)\boldsymbol{\mu}_i^{(\ell-1)} \qquad \text{expand the sum}\\
&= \sum_{j:A_{ij}=0}\mathbb{E}[\alpha_{ij}^{\mathcal{G}'}]\mathbf{m}_{ij}^{(\ell-1)} - Z_i\left(\frac{1}{Z_i}\sum_{j:A_{ij}=0}\mathbb{E}[\alpha_{ij}^{\mathcal{G}'}]\mathbf{m}_{ij}^{(\ell-1)}\right) \qquad \text{definition of } Z_i \text{ and } \boldsymbol{\mu}_i^{(\ell-1)}\\
&= \mathbf{0}.
\end{aligned}$$

$\square$

*Proposition 4.1* (Expectation-Preservation in RADE-OF). Under the RADE perturbation (6), any weighted-sum aggregation (Def. 3.1) corrected by the RADE-OF rule (8) is expectation-preserving (Def. 3.2).

*Proof.* Fix a node $i$ and layer $\ell$. The corrected training-time aggregation is $\widetilde{\mathbf{a}}_i^{(\mathcal{G}',\ell)} = \sum_{j\neq i}\alpha_{ij}^{\mathcal{G}'}\widetilde{\mathbf{m}}_{ij}^{(\ell-1)}$. Splitting the expectation into existing-edge and non-edge terms gives

$$\mathbb{E}\left[\widetilde{\mathbf{a}}_i^{(\mathcal{G}',\ell)}\right] = \mathbb{E}\left[\sum_{j:A_{ij}=1}\alpha_{ij}^{\mathcal{G}'}\widetilde{\mathbf{m}}_{ij}^{(\ell-1)}\right] + \mathbb{E}\left[\sum_{j:A_{ij}=0}\alpha_{ij}^{\mathcal{G}'}\widetilde{\mathbf{m}}_{ij}^{(\ell-1)}\right].$$

For existing edges, based on Lemma B.1, we have $\mathbb{E}\left[\sum_{j:A_{ij}=1}\alpha_{ij}^{\mathcal{G}'}\widetilde{\mathbf{m}}_{ij}^{(\ell-1)}\right] = \sum_{j:A_{ij}=1}\alpha_{ij}^{\mathcal{G}}\mathbf{m}_{ij}^{(\ell-1)}$. For non-edges, RADE-OF uses $\widetilde{\mathbf{m}}_{ij}^{(\ell-1)} = \mathbf{m}_{ij}^{(\ell-1)} - \boldsymbol{\mu}_i^{(\ell-1)}$, Thus, by Lemma B.2,

$$\mathbb{E}\left[\sum_{j:A_{ij}=0}\alpha_{ij}^{\mathcal{G}'}\widetilde{\mathbf{m}}_{ij}^{(\ell-1)}\right] = \sum_{j:A_{ij}=0}\mathbb{E}[\alpha_{ij}^{\mathcal{G}'}]\left(\mathbf{m}_{ij}^{(\ell-1)} - \boldsymbol{\mu}_i^{(\ell-1)}\right) = \mathbf{0}.$$

Combining the two parts and using that $\alpha_{ij}^{\mathcal{G}} = 0$ when $A_{ij} = 0$ give us

$$\mathbb{E}\left[\widetilde{\mathbf{a}}_i^{(\mathcal{G}',\ell)}\right] = \sum_{j:A_{ij}=1}\alpha_{ij}^{\mathcal{G}}\mathbf{m}_{ij}^{(\ell-1)} = \sum_{j\neq i}\alpha_{ij}^{\mathcal{G}}\mathbf{m}_{ij}^{(\ell-1)} = \mathbf{a}_i^{(\mathcal{G},\ell)},$$

which proves that RADE-OF is expectation-preserving. $\square$

*Proposition 4.2* (Expectation-Preservation in RADE-OFS). Under the RADE perturbation (6), any weighted-sum aggregation (Def. 3.1) corrected by the RADE-OFS rule (10) is expectation-preserving with respect to the modified inference-time aggregation in Eq. (11).

*Proof.* Fix a node $i$ and layer $\ell$. The corrected training-time aggregation is $\widetilde{\mathbf{a}}_i^{(\mathcal{G}',\ell)} = \sum_{j \neq i} \alpha_{ij}^{\mathcal{G}'} \widetilde{\mathbf{m}}_{ij}^{(\ell-1)}$. Splitting the expectation into existing-edge and non-edge terms gives

$$\mathbb{E}\left[\widetilde{\mathbf{a}}_i^{(\mathcal{G}',\ell)}\right] = \mathbb{E}\left[\sum_{j:A_{ij}=1} \alpha_{ij}^{\mathcal{G}'} \widetilde{\mathbf{m}}_{ij}^{(\ell-1)}\right] + \mathbb{E}\left[\sum_{j:A_{ij}=0} \alpha_{ij}^{\mathcal{G}'} \widetilde{\mathbf{m}}_{ij}^{(\ell-1)}\right].$$

The existing-edge correction in RADE-OFS is the same as in RADE-OF, so based on Lemma B.1 we have $\mathbb{E}\left[\sum_{j:A_{ij}=1} \alpha_{ij}^{\mathcal{G}'} \widetilde{\mathbf{m}}_{ij}^{(\ell-1)}\right] = \sum_{j:A_{ij}=1} \alpha_{ij}^{\mathcal{G}} \mathbf{m}_{ij}^{(\ell-1)}$. For non-edges, RADE-OFS leaves messages uncentered: $\widehat{\mathbf{m}}_{ij}^{(\ell-1)} = \mathbf{m}_{ij}^{(\ell-1)}$. Thus, $\mathbb{E}\left[\sum_{j:A_{ij}=0} \alpha_{ij}^{\mathcal{G}'} \widetilde{\mathbf{m}}_{ij}^{(\ell-1)}\right] = \sum_{j:A_{ij}=0} \mathbb{E}[\alpha_{ij}^{\mathcal{G}'}] \mathbf{m}_{ij}^{(\ell-1)}$. Combining the two parts,

$$\mathbb{E}\left[\widetilde{\mathbf{a}}_i^{(\mathcal{G}',\ell)}\right] = \sum_{j:A_{ij}=1} \alpha_{ij}^{\mathcal{G}} \mathbf{m}_{ij}^{(\ell-1)} + \sum_{j:A_{ij}=0} \mathbb{E}[\alpha_{ij}^{\mathcal{G}'}] \mathbf{m}_{ij}^{(\ell-1)} = \mathbf{a}_i^{(\mathcal{G},\ell)} + \sum_{j:A_{ij}=0} \mathbb{E}[\alpha_{ij}^{\mathcal{G}'}] \mathbf{m}_{ij}^{(\ell-1)} = \widehat{\mathbf{a}}_i^{(\mathcal{G},\ell)}.$$

Thus, RADE-OFS is expectation-preserving with respect to the modified inference-time aggregation in Eq. (11). $\qquad\square$

## C. Coefficient Expectations for Common Weighted-sum Aggregators

The results in Appendix B apply to any weighted-sum aggregator once the coefficients $\alpha_{ij}^{\mathcal{G}}$ and the expectations $\mathbb{E}[\alpha_{ij}^{\mathcal{G}'}]$ are specified. Here, we instantiate these quantities for sum aggregation, GCN-style symmetric degree normalization, and GAT-style attention normalization. In the following, we denote $d_i$ as the degree of node $i$, including a self-loop, and $n$ as the number of nodes.

### (I) Sum aggregation

For sum aggregation, $\alpha_{ij}^{\mathcal{G}} = A_{ij}$, and under the RADE perturbation (Eq. (6)), we have $\mathbb{E}[\alpha_{ij}^{\mathcal{G}'}] = \mathbb{E}[A'_{ij}]$, which is $1-p$ for dropping edges, and $q$ for adding non-edges. For existing edges, the correction factor in RADE-OF and RADE-OFS (see Eqs. (8) and (10)) becomes $\frac{\alpha_{ij}^{\mathcal{G}}}{\mathbb{E}[\alpha_{ij}^{\mathcal{G}'}]} = \frac{1}{1-p}$. Thus, for sum aggregation, the neighbor-side correction reduces to the usual inverted-drop scaling. For non-edges, $\mathbb{E}[\alpha_{ij}^{\mathcal{G}'}] = q$ is constant over all $j$. Hence the weighted mean in Eq. (9) reduces to the uniform non-neighbor mean: $\boldsymbol{\mu}_i^{(\ell-1)} = \frac{1}{n-d_i} \sum_{j:A_{ij}=0} \mathbf{m}_{ij}^{(\ell-1)}$. Therefore, RADE-OF centers added non-edge messages around this uniform mean. RADE-OFS instead retains its expected contribution in the modified inference-time aggregation of $\widehat{\mathbf{a}}_i^{(\mathcal{G},\ell)} = \mathbf{a}_i^{(\mathcal{G},\ell)} + q \sum_{j:A_{ij}=0} \mathbf{m}_{ij}^{(\ell-1)}$.

### (II) Symmetric-degree aggregation

For GCN-style symmetric degree normalization, $\alpha_{ij}^{\mathcal{G}} = \frac{A_{ij}}{\sqrt{d_i d_j}}$. Thus, the required expectation is $\mathbb{E}[\alpha_{ij}^{\mathcal{G}'}] = \mathbb{E}\left[\frac{A'_{ij}}{\sqrt{d'_i d'_j}}\right]$. Since the coefficient is zero whenever $A'_{ij} = 0$,

$$\mathbb{E}[\alpha_{ij}^{\mathcal{G}'}] = \mathbb{P}(A'_{ij} = 1)\mathbb{E}\left[\frac{1}{\sqrt{d'_i d'_j}} \,\middle|\, A'_{ij} = 1\right] = \mathbb{P}(A'_{ij} = 1)\mathbb{E}\left[(d'_i)^{-1/2} \,\middle|\, A'_{ij} = 1\right]\mathbb{E}\left[(d'_j)^{-1/2} \,\middle|\, A'_{ij} = 1\right]. \quad (15)$$

The main challenge for computing $\mathbb{E}[\alpha_{ij}^{\mathcal{G}'}]$ is the expectation term of endpoint degree-normalization $\mathbb{E}\left[(d'_i)^{-1/2} \,\middle|\, A'_{ij} = 1\right]$, for which we use the following delta-approximation method.

**Lemma C.1** (Endpoint degree-normalization approximation). *Fix a pair $(i, j)$ and condition on $A'_{ij} = 1$. Let $\mu_i = \mathbb{E}[d'_i \mid A'_{ij} = 1]$, and $\sigma_i^2 = \mathrm{Var}(d'_i \mid A'_{ij} = 1)$. Then*

$$\mathbb{E}\left[(d'_i)^{-1/2} \,\middle|\, A'_{ij} = 1\right] \approx \mu_i^{-1/2} + \frac{3}{8}\sigma_i^2 \mu_i^{-5/2}.$$

*Proof.* Let $f(x) = x^{-1/2}$. The second-order delta method gives

$$\mathbb{E}[f(d'_i) \mid A'_{ij} = 1] \approx f(\mu_i) + \frac{1}{2}f''(\mu_i)\sigma_i^2.$$

Since $f''(x) = \frac{3}{4}x^{-5/2}$, we obtain

$$\mathbb{E}\left[(d'_i)^{-1/2} \,\middle|\, A'_{ij} = 1\right] \approx \mu_i^{-1/2} + \frac{3}{8}\sigma_i^2\mu_i^{-5/2}.$$

$\square$

Thus, to approximate Eq. (15), it remains only to specify $\mu_i = \mathbb{E}[d'_i \mid A'_{ij} = 1]$ and $\sigma_i^2 = \mathrm{Var}(d'_i \mid A'_{ij} = 1)$ under the two possible cases that the pair $(i,j)$ is in the input graph (i.e., $A_{ij} = 1$) or not (i.e., $A_{ij} \neq 1$).

**Case $A_{ij} = 1$.** Conditional on $A'_{ij} = 1$, the self-loop and the retained edge $(i,j)$ contribute 2 to $d'_i$. Among the remaining nodes $k \neq i, j$, node $i$ has $d_i - 2$ input-graph neighbors (excluding self and node $j$) and $n - d_i$ input-graph non-neighbors. Hence, $\mu_i = 2 + (d_i - 2)(1 - p) + (n - d_i)q$, and $\sigma_i^2 = (d_i - 2)(1 - p)p + (n - d_i)q(1 - q)$. The terms for $\mu_j$ and $\sigma_j^2$ are similar. Given these terms and since $\mathbb{P}(A'_{ij} = 1) = 1 - p$, we can now approximate:

$$\mathbb{E}[\alpha_{ij}^{\mathcal{G}'}] \approx (1 - p)\left(\mu_i^{-1/2} + \frac{3}{8}\sigma_i^2\mu_i^{-5/2}\right)\left(\mu_j^{-1/2} + \frac{3}{8}\sigma_j^2\mu_j^{-5/2}\right).$$

Thus, for existing edges, the correction factor is $\frac{\alpha_{ij}^{\mathcal{G}}}{\mathbb{E}[\alpha_{ij}^{\mathcal{G}'}]} \approx \frac{1/\sqrt{d_i d_j}}{\mathbb{E}[\alpha_{ij}^{\mathcal{G}'}]}$. Unlike sum aggregation, this correction is pair-dependent because the expected perturbed normalization depends on the endpoint degrees.

**Case $A_{ij} = 0$.** Conditional on $A'_{ij} = 1$, the self-loop and the added edge $(i,j)$ contribute 2 to $d'_i$. Among the remaining nodes $k \neq i, j$, node $i$ has $d_i - 1$ input-graph neighbors (excluding itself) and $n - 1 - d_i$ input-graph non-neighbors excluding self and node $j$. Hence, $\mu_i = 2 + (d_i - 1)(1 - p) + (n - 1 - d_i)q$, and $\sigma_i^2 = (d_i - 1)(1 - p)p + (n - 1 - d_i)q(1 - q)$. The terms for $\mu_j$ and $\sigma_j 2$ are similar. Given these terms, and since $\mathbb{P}(A'_{ij} = 1) = q$, we can approximate

$$\mathbb{E}[\alpha_{ij}^{\mathcal{G}'}] \approx q\left(\mu_i^{-1/2} + \frac{3}{8}\sigma_i^2\mu_i^{-5/2}\right)\left(\mu_j^{-1/2} + \frac{3}{8}\sigma_j^2\mu_j^{-5/2}\right).$$

For RADE-OF, the non-edge expectations define the weighted centering mean in Eq. (9). Unlike sum aggregation, this centering is not uniform over non-neighbors: under GCN normalization, $\mathbb{E}[\alpha_{ij}^{\mathcal{G}'}]$ depends on the perturbed degree moments of both endpoints. Thus, the RADE-OF centering mean is computed by substituting the non-edge approximation into Eq. (9). RADE-OFS uses the same coefficient expectations in its modified inference-time aggregation.

**(III) Attention aggregation**

For GAT-style attention aggregation, the coefficient on graph $\mathcal{G}$ can be written as $\alpha_{ij}^{\mathcal{G}} = \frac{A_{ij}\exp(\beta_{ij})}{\sum_{k \neq i} A_{ik}\exp(\beta_{ik})}$, where $\beta_{ij}$ is the raw attention score before softmax. Conditional on $\mathbf{H}^{(\ell-1)}$, the raw scores are fixed. Thus, on the perturbed graph, $\alpha_{ij}^{\mathcal{G}'} = \frac{A'_{ij}w_{ij}}{\sum_{k \neq i} A'_{ik}w_{ik}}$ with $w_{ik} = \exp(\beta_{ik})$ being constant. Since $\alpha_{ij}^{\mathcal{G}'} = 0$ whenever $A'_{ij} = 0$, we can write

$$\mathbb{E}[\alpha_{ij}^{\mathcal{G}'}] = \mathbb{P}(A'_{ij} = 1)w_{ij}\mathbb{E}\left[\frac{1}{w_{ij} + S_{ij}} \,\middle|\, A'_{ij} = 1\right], \tag{16}$$

where $S_{ij} = \sum_{k \neq i,j} A'_{ik}w_{ik}$. Now, we focus on approximating the expectation term through the combination of delta-expansion and sampling methods.

**Lemma C.2** (Attention-denominator approximation). *Let $S_{ij} = \sum_{k \neq i,j} A'_{ik}w_{ik}$, one can approximate*

$$\mathbb{E}\left[\frac{1}{w_{ij} + S_{ij}}\right] \approx \frac{1}{w_{ij} + \mu_{ij}} + \frac{\sigma_{ij}^2}{(w_{ij} + \mu_{ij})^3},$$

*where $\mu_{ij} = (1 - p)\sum_{\substack{k \neq i,j \\ A_{ik}=1}} w_{ik} + q\sum_{\substack{k \neq i,j \\ A_{ik}=0}} w_{ik}$, and $\sigma_{ij}^2 = p(1-p)\sum_{\substack{k \neq i,j \\ A_{ik}=1}} w_{ik}^2 + q(1-q)\sum_{\substack{k \neq i,j \\ A_{ik}=0}} w_{ik}^2$.*

*Proof.* For $f(x) = (w_{ij} + x)^{-1}$, the second-order delta method gives $\mathbb{E}[f(S_{ij})] \approx f(\mu_{ij}) + \frac{1}{2}f''(\mu_{ij})\sigma_{ij}^2$, where $\mu_{ij} = \mathbb{E}[S_{ij}]$ and $\sigma_{ij}^2 = \mathrm{Var}(S_{ij})$. Since $f''(x) = 2(w_{ij} + x)^{-3}$, the approximation is derived. $\square$

Applying this approximation in Eq. (16), and considering $\mathbb{P}(A'_{ij} = 1)$ for add and drop cases, we obtain

$$
\mathbb{E}[\alpha_{ij}^{\mathcal{G}'}] \approx
\begin{cases}
(1-p)w_{ij}\left(\frac{1}{w_{ij}+\mu_{ij}} + \frac{\sigma_{ij}^2}{(w_{ij}+\mu_{ij})^3}\right), & A_{ij} = 1, \\[2ex]
qw_{ij}\left(\frac{1}{w_{ij}+\mu_{ij}} + \frac{\sigma_{ij}^2}{(w_{ij}+\mu_{ij})^3}\right), & A_{ij} = 0.
\end{cases}
\tag{17}
$$

For existing edges, the RADE-OF and RADE-OFS correction factors become $\alpha_{ij}^{\mathcal{G}}/\mathbb{E}[\alpha_{ij}^{\mathcal{G}'}]$. Thus, as in GCN-style normalization, the correction is pair-dependent. For non-edges, the same expectation terms define the weighted centering mean $\boldsymbol{\mu}_i^{(\ell-1)}$ in RADE-OF, while RADE-OFS retains the expected non-edge contribution in the modified inference-time aggregation.

The computational bottleneck in Eq. (17) is the computation of pair-specific $\mu_{ij}, \sigma_{ij}^2$; specifically, the bottleneck is in the computation of their non-edge part. In practice, we compute the input-edge terms exactly and estimate only the non-edge terms by sampling. We sample once per target node $i$, not once per pair $(i,j)$. We uniformly sample $s$ non-neighbors of node $i$, denoted by $r_1, \ldots, r_s$, through rejection sampling, which rejects existing edges and the self-node $i$. Since the same sample is reused for all pairs $(i,j)$, we do not reject the candidate $j$ during sampling. Instead, we first estimate $\widehat{\mu}_i = (1-p)\sum_{\substack{k\neq i \\ A_{ik}=1}} w_{ik} + q\frac{|\{k:k\neq i,\, A_{ik}=0\}|}{s}\sum_{t=1}^s w_{ir_t}$, and $\widehat{\sigma}_i^2 = p(1-p)\sum_{\substack{k\neq i \\ A_{ik}=1}} w_{ik}^2 + q(1-q)\frac{|\{k:k\neq i,\, A_{ik}=0\}|}{s}\sum_{t=1}^s w_{ir_t}^2$. Then, for each pair $(i,j)$, we form the pair-excluded estimates by subtraction: $\widehat{\mu}_{ij} = \widehat{\mu}_i - (1-p)w_{ij}$ for $A_{ij} = 1$, $\widehat{\mu}_{ij} = \widehat{\mu}_i - qw_{ij}$ for $A_{ij} = 0$, and $\widehat{\sigma}_{ij}^2 = \widehat{\sigma}_i^2 - p(1-p)w_{ij}^2$ for $A_{ij} = 1$, $\widehat{\sigma}_{ij}^2 = \widehat{\sigma}_i^2 - q(1-q)w_{ij}^2$ for $A_{ij} = 0$. Substituting these estimates into the approximation above gives the sampled version of $\mathbb{E}[\alpha_{ij}^{\mathcal{G}'}]$. The same sampling idea is used for the non-edge centering term in RADE-OF and the modified inference-time non-edge contribution in RADE-OFS.

## D. Proof of Proposition 4.3

**Proposition 4.3** (Logit perturbation variance). Assume that, for each node $i$, the stochastic aggregation coefficients $\{\alpha_{ij}^{\mathcal{G}'}\}_{j\neq i}$ are mutually independent across $j$. Then $\mathrm{Var}(\delta_i) = \sum_{j\neq i} \widetilde{m}_{ij}^2 \, \mathrm{Var}\left(\alpha_{ij}^{\mathcal{G}'}\right)$. Multiclass results hold coordinate-wise.

*Proof.* Condition on $\mathbf{H}^{(L-1)}$, so that the scalar message contributions $\{\widetilde{m}_{ij}\}_{j\neq i}$ are deterministic with respect to the augmentation randomness. At the final layer, the perturbed logit can be written as $\widetilde{s}_i = \sum_{j\neq i} \alpha_{ij}^{\mathcal{G}'} \widetilde{m}_{ij}$, where $\delta_i = \widetilde{s}_i - s_i$. Since $s_i$ is deterministic under the conditioning,

$$
\mathrm{Var}(\delta_i) = \mathrm{Var}(\widetilde{s}_i - s_i) = \mathrm{Var}(\widetilde{s}_i) = \mathrm{Var}\left(\sum_{j\neq i} \alpha_{ij}^{\mathcal{G}'} \widetilde{m}_{ij}\right) = \sum_{j\neq i} \mathrm{Var}\left(\alpha_{ij}^{\mathcal{G}'} \widetilde{m}_{ij}\right) = \sum_{j\neq i} \widetilde{m}_{ij}^2 \mathrm{Var}\left(\alpha_{ij}^{\mathcal{G}'}\right)
$$

For multiclass classification with a linear head, the same argument applies to each logit coordinate. For class $c$, let $\widetilde{s}_{i,c} = \sum_{j\neq i} \alpha_{ij}^{\mathcal{G}'} \widetilde{m}_{ij,c}$, where $\delta_{i,c} = \widetilde{s}_{i,c} - s_{i,c}$. Then $\mathrm{Var}(\delta_{i,c}) = \sum_{j\neq i} \widetilde{m}_{ij,c}^2 \mathrm{Var}\left(\alpha_{ij}^{\mathcal{G}'}\right)$. $\qquad\square$

## E. Logit Variance for RADE-OF and RADE-OFS

Proposition 4.3 shows that, conditioned on $\mathbf{H}^{(L-1)}$, the logit perturbation variance is $\mathrm{Var}(\delta_i) = \sum_{j\neq i} \widetilde{m}_{ij}^2 \mathrm{Var}\left(\alpha_{ij}^{\mathcal{G}'}\right)$, where $\widetilde{m}_{ij}$ is the scalar corrected contribution of message $j$ to $i$ for the final layer. Thus, the computation of $\mathrm{Var}(\delta_i)$ requires computing $\mathrm{Var}(\alpha_{ij}^{\mathcal{G}'})$ for any chosen aggregation rule. Below, we derive this term for sum aggregation (in GIN), GCN-style symmetric normalization, and GAT-style attention aggregation. When possible, we also simplify the logit variance expression for different aggregation rules by first decomposing it into two sums corresponding to existing edges and non-edges:

$$
\mathrm{Var}(\delta_i) = \sum_{j:A_{ij}=1} \widetilde{m}_{ij}^2 \mathrm{Var}\left(\alpha_{ij}^{\mathcal{G}'}\right) + \sum_{j:A_{ij}=0} \widetilde{m}_{ij}^2 \mathrm{Var}\left(\alpha_{ij}^{\mathcal{G}'}\right),
$$

**(I) Sum aggregation**

For sum aggregation, $\alpha_{ij}^{\mathcal{G}} = A_{ij}$. Under the RADE perturbation (Eq. (6)), we have $\mathbb{E}[\alpha_{ij}^{\mathcal{G}'}] = \mathbb{E}[A_{ij}']$, which is $1 - p$ for dropping edges, and $q$ for adding non-edges. Then, $\mathrm{Var}\left(\alpha_{ij}^{\mathcal{G}'}\right) = p(1 - p)$ for $A_{ij} = 1$ and $q(1 - q)$ for $A_{ij} = 0$. For the existing-edge term, both RADE-OF and RADE-OFS give

$$\sum_{j:A_{ij}=1} \widetilde{m}_{ij}^2 \mathrm{Var}\left(\alpha_{ij}^{\mathcal{G}'}\right) = \sum_{j:A_{ij}=1} \frac{\left(\alpha_{ij}^{\mathcal{G}}\right)^2 m_{ij}^2}{\left(\mathbb{E}[\alpha_{ij}^{\mathcal{G}'}]\right)^2} \mathrm{Var}\left(\alpha_{ij}^{\mathcal{G}'}\right) = \sum_{j:A_{ij}=1} \frac{m_{ij}^2}{(1-p)^2} p(1-p) = \frac{p}{1-p} \sum_{j:A_{ij}=1} m_{ij}^2.$$

For non-edges, $\mathbb{E}[\alpha_{ij}^{\mathcal{G}'}] = q$ is constant over all $j$ such that $A_{ij} = 0$. Thus, the RADE-OF weighted mean reduces to the uniform non-neighbor mean: $\mu_i = \frac{1}{n-d_i} \sum_{j:A_{ij}=0} m_{ij}$. Then, the addition-induced term is $q(1 - q) \sum_{j:A_{ij}=0} (m_{ij} - \mu_i)^2$ for RADE-OF, and $q(1 - q) \sum_{j:A_{ij}=0} m_{ij}^2$ for RADE-OFS. Thus, under sum aggregation,

$$\mathrm{Var}(\delta_i) = \frac{p}{1-p} \sum_{j:A_{ij}=1} m_{ij}^2 + q(1-q) \sum_{j:A_{ij}=0} (m_{ij} - \mu_i)^2$$

for RADE-OF, while

$$\mathrm{Var}(\delta_i) = \frac{p}{1-p} \sum_{j:A_{ij}=1} m_{ij}^2 + q(1-q) \sum_{j:A_{ij}=0} m_{ij}^2$$

for RADE-OFS. The difference between the two variants is exactly the centering of the non-edge messages. Since $\sum_{j:A_{ij}=0} m_{ij}^2 = \sum_{j:A_{ij}=0} (m_{ij} - \mu_i)^2 + (n - d_i) \mu_i^2$, RADE-OFS has an additional non-edge variance contribution of $q(1 - q)(n - d_i) \mu_i^2$ relative to RADE-OF under sum aggregation.

**(II) Symmetric degree normalization**

For GCN-style symmetric normalization, $\alpha_{ij}^{\mathcal{G}} = \frac{A_{ij}}{\sqrt{d_i d_j}}$. The coefficient expectations $\mathbb{E}[\alpha_{ij}^{\mathcal{G}'}]$ were derived in Appendix C. To obtain the variance terms, we need the second moment $\mathbb{E}[(\alpha_{ij}^{\mathcal{G}'})^2]$, since $\mathrm{Var}\left(\alpha_{ij}^{\mathcal{G}'}\right) = \mathbb{E}\left[(\alpha_{ij}^{\mathcal{G}'})^2\right] - \left(\mathbb{E}[\alpha_{ij}^{\mathcal{G}'}]\right)^2$. As $(A_{ij}')^2 = A_{ij}'$, $(\alpha_{ij}^{\mathcal{G}'})^2 = \frac{A_{ij}'}{d_i' d_j'}$. Thus, conditioning on $A_{ij}' = 1$ gives

$$\mathbb{E}\left[(\alpha_{ij}^{\mathcal{G}'})^2\right] = \mathbb{P}(A_{ij}' = 1)\mathbb{E}\left[(d_i')^{-1} \,\middle|\, A_{ij}' = 1\right]\mathbb{E}\left[(d_j')^{-1} \,\middle|\, A_{ij}' = 1\right].$$

The factorization uses the same conditional-independence argument as in Appendix C. We use the following delta-method approximation for each endpoint inverse-degree term.

**Lemma E.1** (Endpoint inverse-degree approximation). *Fix a pair $(i, j)$ and condition on $A_{ij}' = 1$. Let $\mu_i = \mathbb{E}[d_i' \mid A_{ij}' = 1]$, and $\sigma_i^2 = \mathrm{Var}(d_i' \mid A_{ij}' = 1)$. Then $\mathbb{E}\left[(d_i')^{-1} \,\middle|\, A_{ij}' = 1\right] \approx \mu_i^{-1} + \sigma_i^2 \mu_i^{-3}$.*

*Proof.* Let $g(x) = x^{-1}$. The second-order delta method gives

$$\mathbb{E}[g(d_i') \mid A_{ij}' = 1] \approx g(\mu_i) + \frac{1}{2}g''(\mu_i)\sigma_i^2 = \mu_i^{-1} + \frac{1}{2}\left(2\mu_i^{-3}\right)\sigma_i^2 = \mu_i^{-1} + \sigma_i^2\mu_i^{-3}.$$

$\square$

For an existing edge $A_{ij} = 1$, we use the case $A_{ij} = 1$ conditional moments $\mu_i, \sigma_i^2, \mu_j, \sigma_j^2$ from Appendix C. Since $\mathbb{P}(A_{ij}' = 1) = 1 - p$, $\mathbb{E}\left[(\alpha_{ij}^{\mathcal{G}'})^2\right] \approx (1-p)\left(\mu_i^{-1} + \sigma_i^2\mu_i^{-3}\right)\left(\mu_j^{-1} + \sigma_j^2\mu_j^{-3}\right)$, and $\mathrm{Var}\left(\alpha_{ij}^{\mathcal{G}'}\right) \approx (1-p)\left(\mu_i^{-1} + \sigma_i^2\mu_i^{-3}\right)\left(\mu_j^{-1} + \sigma_j^2\mu_j^{-3}\right) - \left(\mathbb{E}[\alpha_{ij}^{\mathcal{G}'}]\right)^2$.

For a non-edge $A_{ij} = 0$, we use the Case $A_{ij} = 0$ conditional moments from Appendix C. Since $\mathbb{P}(A_{ij}' = 1) = q$, $\mathbb{E}\left[(\alpha_{ij}^{\mathcal{G}'})^2\right] \approx q\left(\mu_i^{-1} + \sigma_i^2\mu_i^{-3}\right)\left(\mu_j^{-1} + \sigma_j^2\mu_j^{-3}\right)$, $\mathrm{Var}\left(\alpha_{ij}^{\mathcal{G}'}\right) \approx q\left(\mu_i^{-1} + \sigma_i^2\mu_i^{-3}\right)\left(\mu_j^{-1} + \sigma_j^2\mu_j^{-3}\right) - \left(\mathbb{E}[\alpha_{ij}^{\mathcal{G}'}]\right)^2$.

Substituting this coefficient variance into the generic RADE-OF and RADE-OFS logit-variance forms above gives the GCN variance instantiations.

**(III) Attention normalization**

For GAT-style attention aggregation, the coefficient on graph $\mathcal{G}$ can be written as $\alpha_{ij}^{\mathcal{G}} = \frac{A_{ij}\exp(\beta_{ij})}{\sum_{k\neq i} A_{ik}\exp(\beta_{ik})}$, where $\beta_{ij}$ is the raw attention score before softmax. Conditional on $\mathbf{H}^{(L-1)}$, the raw scores are fixed. Then, on the perturbed graph, $\alpha_{ij}^{\mathcal{G}'} = \frac{A'_{ij} w_{ij}}{\sum_{k\neq i} A'_{ik} w_{ik}}$ with $w_{ik} = \exp(\beta_{ik})$ being constant. As in Appendix C, fix a pair $(i, j)$, condition on $A'_{ij} = 1$, and write $S_{ij} = \sum_{k\neq i,j} A'_{ik} w_{ik}$. Let $\mu_{ij} = (1-p)\sum_{\substack{k\neq i,j \\ A_{ik}=1}} w_{ik} + q\sum_{\substack{k\neq i,j \\ A_{ik}=0}} w_{ik}$, and $\sigma_{ij}^2 = p(1-p)\sum_{\substack{k\neq i,j \\ A_{ik}=1}} w_{ik}^2 + q(1-q)\sum_{\substack{k\neq i,j \\ A_{ik}=0}} w_{ik}^2$. The first moment $\mathbb{E}[\alpha_{ij}^{\mathcal{G}'}]$ is given by Eq. (17). To compute the coefficient variance, we also need the second moment:

$$\mathbb{E}[(\alpha_{ij}^{\mathcal{G}'})^2] = \mathbb{P}(A'_{ij} = 1) w_{ij}^2 \mathbb{E}\left[\frac{1}{(w_{ij} + S_{ij})^2} \mid A'_{ij} = 1\right], \tag{18}$$

where we use the following lemma for its approximation.

**Lemma E.2** (Attention second-moment approximation). *With $S_{ij}$, $\mu_{ij}$, and $\sigma_{ij}^2$ defined above,*

$$\mathbb{E}\left[\frac{1}{(w_{ij} + S_{ij})^2}\right] \approx \frac{1}{(w_{ij} + \mu_{ij})^2} + \frac{3\sigma_{ij}^2}{(w_{ij} + \mu_{ij})^4}.$$

*Proof.* For $g(x) = (w_{ij} + x)^{-2}$, the second-order delta method gives $\mathbb{E}[g(S_{ij})] \approx g(\mu_{ij}) + \frac{1}{2}g''(\mu_{ij})\sigma_{ij}^2$, where $\mu_{ij} = \mathbb{E}[S_{ij}]$, and $\sigma_{ij}^2 = \mathrm{Var}(S_{ij})$. Since $g''(x) = 6(w_{ij} + x)^{-4}$, the result follows. $\square$

Applying Lemma E.2 to Eq. (18), and considering $\mathbb{P}(A'_{ij} = 1)$ for add and drop cases, we obtain

$$\mathbb{E}\left[(\alpha_{ij}^{\mathcal{G}'})^2\right] \approx \begin{cases} (1-p)w_{ij}^2\left(\frac{1}{(w_{ij}+\mu_{ij})^2} + \frac{3\sigma_{ij}^2}{(w_{ij}+\mu_{ij})^4}\right), & A_{ij} = 1, \\ qw_{ij}^2\left(\frac{1}{(w_{ij}+\mu_{ij})^2} + \frac{3\sigma_{ij}^2}{(w_{ij}+\mu_{ij})^4}\right), & A_{ij} = 0. \end{cases}$$

Thus,

$$\mathrm{Var}\left(\alpha_{ij}^{\mathcal{G}'}\right) \approx \mathbb{E}\left[(\alpha_{ij}^{\mathcal{G}'})^2\right] - \left(\mathbb{E}[\alpha_{ij}^{\mathcal{G}'}]\right)^2.$$

The computational bottleneck is again the computation of $\mu_{ij}$ and $\sigma_{ij}^2$, in $\mathrm{Var}(\alpha_{ij}^{\mathcal{G}'})$, especially for their non-edge part. Thus, we compute the input-edge terms exactly and estimate only the non-edge terms by sampling. We uniformly sample $s$ non-neighbors for node $i$, denoted by $r_1, \ldots, r_s$, through rejection sampling, which rejects existing edges and the self-node $i$. Since the same sample is reused for all pairs $(i, j)$, we do not reject the candidate $j$ during sampling. Instead, we first estimate $\widehat{\mu}_i = (1-p)\sum_{\substack{k\neq i \\ A_{ik}=1}} w_{ik} + q\frac{|\{k:k\neq i,\ A_{ik}=0\}|}{s}\sum_{t=1}^{s} w_{ir_t}$, and $\widehat{\sigma}_i^2 = p(1-p)\sum_{\substack{k\neq i \\ A_{ik}=1}} w_{ik}^2 + q(1-q)\frac{|\{k:k\neq i,\ A_{ik}=0\}|}{s}\sum_{t=1}^{s} w_{ir_t}^2$. Then, for each pair $(i, j)$, we form the pair-excluded estimates by subtraction: $\widehat{\mu}_{ij} = \widehat{\mu}_i - (1-p)w_{ij}$ for $A_{ij} = 1$, $\widehat{\mu}_{ij} = \widehat{\mu}_i - qw_{ij}$ for $A_{ij} = 0$, and $\widehat{\sigma}_{ij}^2 = \widehat{\sigma}_i^2 - p(1-p)w_{ij}^2$ for $A_{ij} = 1$, $\widehat{\sigma}_{ij}^2 = \widehat{\sigma}_i^2 - q(1-q)w_{ij}^2$ for $A_{ij} = 0$. Substituting these estimates into the moment approximations above gives the sampled version of $\mathrm{Var}(\alpha_{ij}^{\mathcal{G}'})$. Substituting this coefficient variance into Proposition 4.3 gives the GAT variance instantiation.

The same sampling idea is used for the final non-edge sums in $\mathrm{Var}(\delta_i)$: for RADE-OF, we estimate $\sum_{\substack{j\neq i \\ A_{ij}=0}} (m_{ij} - \mu_i)^2 \mathrm{Var}(\alpha_{ij}^{\mathcal{G}'})$, while for RADE-OFS, we estimate $\sum_{\substack{j\neq i \\ A_{ij}=0}} m_{ij}^2 \mathrm{Var}(\alpha_{ij}^{\mathcal{G}'})$.

## F. Proof of Proposition 4.4

*Proposition 4.4* (Restricted interchangeability of Drop-only and Add-only). For sum aggregation, Drop-only and Add-only only subsume each other by matching their node-wise variances under a restrictive setting in which there should exist a constant $\kappa > 0$ such that $\sum_{j:A_{ij}=0} \widetilde{m}_{ij}^2 = \kappa \sum_{j:A_{ij}=1} m_{ij}^2$ for every node $i$.

*Proof.* Under sum aggregation, the RADE variance decomposition gives the node-wise variances for the two pure strategies. Drop-only, obtained by setting $q = 0$, gives $\mathrm{Var}(\delta_i)\big|_{q=0} = \frac{p}{1-p} \sum_{j:A_{ij}=1} m_{ij}^2$. Add-only, obtained by setting the deletion rate to zero, gives $\mathrm{Var}(\delta_i)\big|_{p=0} = q(1-q) \sum_{j:A_{ij}=0} \widetilde{m}_{ij}^2$. For exact node-by-node matching with single global rates, one requires that these two quantities be equal for every node $i$: $\frac{p}{1-p} \sum_{j:A_{ij}=1} m_{ij}^2 = q(1-q) \sum_{j:A_{ij}=0} \widetilde{m}_{ij}^2$. For a single node, this equality may be satisfied by choosing an appropriate rate. However, there is a restriction that the same rates must match all nodes. For two nodes $i$ and $k$, matching requires

$$\frac{\sum_{j:A_{ij}=0} \widetilde{m}_{ij}^2}{\sum_{j:A_{ij}=1} m_{ij}^2} = \frac{\frac{p}{1-p}}{q(1-q)} = \frac{\sum_{j:A_{kj}=0} \widetilde{m}_{kj}^2}{\sum_{j:A_{kj}=1} m_{kj}^2},$$

whenever the edge-message statistics in the denominators are nonzero. Thus, the ratio between the non-edge and edge message statistics must be the same for both nodes. Extending this argument to all nodes, exact global matching is possible only if there exists a constant $\kappa > 0$ such that $\sum_{j:A_{ij}=0} \widetilde{m}_{ij}^2 = \kappa \sum_{j:A_{ij}=1} m_{ij}^2$ for every node $i$. This proportionality condition is restrictive because Drop-only message statistic for each node $i$ is impacted by node $i$'s existing edges, whereas its Add-only counterpart depends on a distinct set of non-edges. Even these sets change from one node to another.

$\square$

## G. Adaptive $(p, q)$ via Gradient-Norm Balancing

We describe the procedure to adapt the RADE augmentation rates $(p, q)$ during training. The goal is to choose perturbation rates whose variance-based regularization effect has a gradient scale comparable to that of the supervised objective, while avoiding excessive graph densification from edge addition. The training procedure is summarized in Algorithm 1.

Let $\theta_s$ denote the shared trainable parameters. For a mini-batch $B$, let $L_{\mathrm{data}}(B)$ denote the supervised objective on batch $B$. We measure the supervised gradient norm by $G_{\mathrm{data}}^B = \|\nabla_{\theta_s} L_{\mathrm{data}}(B)\|_2$. For a candidate pair $(p, q)$, we define the RADE regularization proxy as $R(B, p, q) = \frac{1}{2} \sum_{i \in B} z_i(1 - z_i) \mathrm{Var}(\delta_i)$, with $z_i = \sigma(s_i)$, $s_i$ the logit from the input-graph forward pass, and $\mathrm{Var}(\delta_i)$ the RADE logit-variance expression under rates $(p, q)$. For multi-class classification, with $\mathbf{p}_i = \mathrm{softmax}(\mathbf{s}_i)$, we use the coordinate-wise analogue $R(B, p, q) = \frac{1}{2} \sum_{i \in B} \sum_{c=1}^C p_{i,c}(1 - p_{i,c}) \mathrm{Var}(\delta_{i,c})$. The corresponding regularization gradient norm is $G_{\mathrm{reg}}^B(p, q) = \|\nabla_{\theta_s} R(B, p, q)\|_2$. We update $(p, q)$ by minimizing the mini-batch objective

$$\mathcal{J}(p, q) = \left[\log\left(\frac{G_{\mathrm{reg}}^B(p, q) + \epsilon}{G_{\mathrm{data}}^B + \epsilon}\right)\right]^2 + \lambda\left(\frac{q}{D(\mathcal{G})}\right)^2.$$

The first term matches the regularization and supervised gradient magnitudes in relative scale. The stabilizer $\epsilon > 0$ prevents numerical instability when either norm is small. The second term penalizes edge addition relative to the input-graph density $D(\mathcal{G})$. This is important because $q$ is applied over non-edges, so even a small raw value of $q$ can add many edges to sparse graphs. The parameter $\lambda$ controls this trade-off: $\lambda = 0$ recovers pure gradient-norm matching, while larger $\lambda$ yields more conservative edge addition. We fix $\lambda$ at the task level, using $\lambda = 1$ for node classification and $\lambda = 0.1$ for graph classification, rather than tuning it per dataset.

In practice, after each mini-batch, we take one Adam step on $\mathcal{J}(p, q)$ with respect to the rate variables. The updated rates are then used to sample RADE perturbations for the next mini-batch. In full-batch training, there is only one batch per epoch, so the same update rule reduces to one rate update per epoch.

**Density-normalized addition coordinate.** To make the addition rate comparable across graphs with different densities, we optimize the density-normalized coordinate $\rho = \frac{q}{D(\mathcal{G})}$, where $D(\mathcal{G})$ is the graph density. Thus, Adam updates act on relative edge addition rather than raw non-edge probability. This is only a change of optimization coordinate: the RADE variance formulas and the edge-sampling procedure are still evaluated using the actual addition probability $q$. The penalty term $\lambda(q/D(\mathcal{G}))^2 = \lambda\rho^2$ therefore directly discourages excessive densification relative to the input graph density. After each update, we project $(p, \rho)$ to the feasible range and map back to $q = D(\mathcal{G})\rho$ for sampling.

**Detached-variance convention.** To keep the GradNorm computation lightweight, we use a detached-variance convention when computing $G_{\mathrm{reg}}^B(p, q)$. The message statistics entering $\mathrm{Var}(\delta_i)$ are computed from a forward pass on the input graph and are treated as fixed when differentiating $R(B, p, q)$ with respect to $\theta_s$. Thus, gradients in $\nabla_{\theta_s} R(B, p, q)$ flow through the uncertainty weights $z_i(1 - z_i)$, or $p_{i,c}(1 - p_{i,c})$ in the multi-class case, while the variance term acts as a nonnegative coefficient controlled by $(p, q)$. The variance expression remains a function of $(p, q)$ for the rate update.

---

**Algorithm 1** RADE Training with Mini-batch PQ-GradNorm Updates

---

**Require:** training loader producing mini-batches $B$; epochs $E$
**Require:** model parameters $\theta$ with shared parameters $\theta_s$; model optimizer $\mathrm{Opt}_\theta$
**Require:** rate optimizer $\mathrm{Opt}_{p,\rho}$ for $(p,\rho)$
**Require:** initial rates $(p,\rho)$; graph density $D(\mathcal{G})$; penalty weight $\lambda$; numerical constant $\epsilon$
1: **for** $e = 1,\ldots,E$ **do**
2:     **for** mini-batch $B$ in the training loader **do**
3:         set $q = D(\mathcal{G})\rho$
4:         sample RADE perturbation on the current batch using current $(p,q)$
5:         compute supervised loss $L_{\mathrm{data}}(B)$
6:         update model parameters: $\theta \leftarrow \mathrm{Opt}_\theta(\theta, \nabla_\theta L_{\mathrm{data}}(B))$
7:         compute $G^B_{\mathrm{data}}$ and $G^B_{\mathrm{reg}}(p,q)$, with $q = D(\mathcal{G})\rho$
8:         form

$$\mathcal{J}(p,\rho) = \left[\log\left(\frac{G^B_{\mathrm{reg}}(p, D(\mathcal{G})\rho) + \epsilon}{G^B_{\mathrm{data}} + \epsilon}\right)\right]^2 + \lambda\rho^2$$

9:         take one Adam step on $\mathcal{J}(p,\rho)$ with respect to $(p,\rho)$
10:       project $(p,\rho)$ to the feasible range and set $q = D(\mathcal{G})\rho$
11:     **end for**
12: **end for**

---

**GIN, GCN, and GAT evaluation.** For GIN-style sum aggregation, the variance proxy separates into deletion and addition components, and for a fixed batch, $R(B,p,q) = \frac{p}{1-p}R_{\mathrm{drop}}(B) + q(1-q)R_{\mathrm{add}}(B)$. Once the two base gradients $\nabla_{\theta_s}R_{\mathrm{drop}}(B)$ and $\nabla_{\theta_s}R_{\mathrm{add}}(B)$ are computed, the regularization gradient norm for any $(p,q)$ can be evaluated analytically:

$$\begin{aligned}
\left(G^B_{\mathrm{reg}}(p,q)\right)^2 &= \left\|\frac{p}{1-p}\nabla_{\theta_s}R_{\mathrm{drop}}(B) + q(1-q)\nabla_{\theta_s}R_{\mathrm{add}}(B)\right\|_2^2 \\
&= \left(\frac{p}{1-p}\right)^2 \|\nabla_{\theta_s}R_{\mathrm{drop}}(B)\|_2^2 + \left(q(1-q)\right)^2 \|\nabla_{\theta_s}R_{\mathrm{add}}(B)\|_2^2 \\
&\quad + 2\frac{p}{1-p}q(1-q)\langle\nabla_{\theta_s}R_{\mathrm{drop}}(B), \nabla_{\theta_s}R_{\mathrm{add}}(B)\rangle.
\end{aligned}$$

This makes the GIN case inexpensive: after computing the two base gradients once, the regularization gradient norm for any candidate $(p,q)$ is obtained by an algebraic combination, without an additional backward pass.

For GCN and GAT, this fixed decomposition is unavailable. In GCN, $(p,q)$ changes the degree-normalization moments entering $\mathrm{Var}(\delta_i)$. In GAT, $(p,q)$ changes the attention-denominator moments, and thus the coefficient expectations and variances. Therefore, for each current or candidate $(p,q)$ considered by the rate optimizer, we recompute the corresponding variance expression, form $R(B,p,q)$ under the same detached-variance convention, and differentiate it to obtain $G^B_{\mathrm{reg}}(p,q)$.

**Graph classification.** Graph classification follows the same principle, except that the supervised logit is graph-level. RADE perturbs node representations through message passing, and the readout pools these perturbations into a graph-level logit perturbation. Thus, $\delta_i$ denotes the perturbation of the graph-level logit for graph $i$, after readout and prediction, rather than a node-level logit perturbation. With linear readout and a linear prediction head, this perturbation is obtained by pooling the node-level perturbation contributions; under sum pooling, this corresponds to summing node-level variance contributions.

**Mini-batch semantics.** For neighbor-sampling node classification, the supervised loss is taken over the labeled seed nodes, while the RADE variance proxy is constructed from the corresponding sampled computation graph; the GradNorm objective is then evaluated on those same seed nodes. For graph classification, complement-based quantities, such as non-edge centering statistics, are computed separately within each graph in the batch, not across different graphs.

## H. Additional Experiments

We report experimental results under GIN and GAT backbones for node- and graph-classification tasks.

*Table 7.* Node classification results (mean±std over five runs) with GIN and GAT backbones. Accuracy (%) on all datasets except Flickr with micro-F1 (%). Best in bold, second underlined. Green and red shading show improvement and degradation over the backbone.

| Method | Cora | CiteSeer | PubMed | CS | Computer | Physics | Flickr | ogbn-arxiv |
|---|---|---|---|---|---|---|---|---|
| | | | | **Backbone: GIN** | | | | |
| GIN | 77.96±2.66 | 64.04±2.64 | 75.88±0.19 | 91.79±0.38 | 87.58±3.15 | 95.77±0.02 | 52.98±0.19 | 70.46±0.19 |
| Dropout | 76.66±1.19 | 64.44±1.32 | 75.12±1.32 | 92.88±0.28 | **90.76±0.17** | 96.01±0.12 | 53.25±0.13 | 70.77±0.42 |
| DropEdge | 77.56±1.77 | 65.02±1.73 | 76.54±0.71 | 91.97±0.11 | 89.64±0.52 | 95.96±0.07 | 52.74±0.30 | 71.00±0.32 |
| DropNode | 77.20±0.80 | 65.66±1.01 | 75.88±0.66 | 91.73±0.20 | 89.72±0.32 | 95.97±0.15 | 52.83±1.23 | 71.49±0.24 |
| DropMessage | 78.20±1.83 | 64.70±1.70 | 77.14±0.70 | 92.64±0.35 | 90.38±0.90 | 96.20±0.09 | 52.98±0.19 | 71.55±0.20 |
| RADE-OF | **78.64±0.68** | 65.94±1.45 | 77.09±0.95 | **93.01±0.14** | 90.50±0.15 | **96.25±0.10** | **53.35±0.08** | **71.85±0.32** |
| RADE-OFS | 78.22±2.17 | 65.69±1.85 | **77.52±1.19** | 92.75±0.25 | 89.82±0.48 | 96.24±0.05 | 53.02±0.15 | 70.85±0.65 |
| | | | | **Backbone: GAT** | | | | |
| GAT | 77.44±1.03 | 65.12±0.80 | 73.46±0.96 | 90.75±0.68 | 88.86±0.48 | 95.90±0.06 | 51.40±0.24 | 69.51±0.42 |
| Dropout | 78.04±1.00 | **66.70±1.26** | 74.78±1.72 | 91.44±0.18 | 89.67±0.28 | 95.95±0.09 | 51.41±0.42 | 70.63±0.10 |
| DropEdge | 78.10±1.06 | 65.44±1.02 | 74.06±1.08 | 91.23±0.34 | 90.13±0.13 | 96.01±0.09 | 51.92±0.16 | 69.72±0.14 |
| DropNode | 79.22±2.29 | 64.34±1.79 | 74.16±0.98 | 91.41±0.39 | 82.70±1.57 | 96.01±0.05 | 51.95±0.10 | 70.18±0.34 |
| DropMessage | 79.72±1.30 | 63.78±1.51 | 75.82±1.45 | 88.44±0.69 | 89.99±0.30 | 95.96±0.05 | 51.30±0.18 | 69.83±0.27 |
| RADE-OF | **80.32±0.78** | 66.34±0.85 | 76.96±0.60 | **92.08±0.13** | 90.40±0.36 | 96.03±0.25 | 51.98±0.21 | 70.98±0.12 |
| RADE-OFS | 80.22±1.44 | 66.10±0.72 | **77.02±1.15** | 92.07±0.24 | 90.24±0.12 | 95.93±0.14 | 51.94±0.15 | 70.66±0.16 |

*Table 8.* Graph classification results (mean±std) with GIN and GAT backbones. Accuracy (%) on TU, ROC-AUC (%) on ogbg-molhiv, and AP (%) on peptides-func. Best in bold and second underlined. Green/red shading shows gains/losses vs. the backbone.

| Method | MUTAG | PROTEINS | IMDB-B | IMDB-M | ogbg-molhiv | peptides-func |
|---|---|---|---|---|---|---|
| | | | **Backbone: GIN** | | | |
| GIN | 84.22 ± 6.12 | 75.19 ± 5.27 | 72.40 ± 2.93 | 48.66 ± 4.40 | 73.84 ± 1.25 | 55.63 ± 1.24 |
| Dropout | 84.41 ± 8.94 | 74.74 ± 3.81 | 71.10 ± 7.68 | 48.73 ± 4.83 | 73.98 ± 1.66 | 55.65 ± 0.82 |
| DropEdge | 85.05 ± 6.28 | 75.14 ± 4.62 | 71.70 ± 3.46 | 47.73 ± 2.68 | 74.02 ± 1.01 | 58.18 ± 0.68 |
| DropNode | 85.02 ± 8.95 | 75.73 ± 4.14 | 72.60 ± 3.13 | 50.13 ± 3.58 | 73.90 ± 1.20 | 55.12 ± 0.72 |
| DropMessage | 84.50 ± 6.69 | 74.39 ± 3.15 | **74.30 ± 3.37** | 47.60 ± 2.93 | 74.18 ± 1.62 | 58.48 ± 0.84 |
| RADE-OF | **85.55 ± 8.19** | 75.82 ± 2.98 | 73.70 ± 3.52 | 50.15 ± 4.62 | 74.30 ± 1.02 | 58.82 ± 0.65 |
| RADE-OFS | 85.32 ± 7.75 | **75.95 ± 3.10** | 74.10 ± 2.80 | **50.32 ± 2.96** | **74.52 ± 1.34** | **59.12 ± 0.91** |
| | | | **Backbone: GAT** | | | |
| GAT | 83.45 ± 5.71 | 75.64 ± 5.12 | 74.10 ± 2.73 | 50.46 ± 2.34 | 68.41 ± 1.43 | 53.16 ± 3.14 |
| Dropout | 83.97 ± 5.99 | 74.56 ± 4.96 | 74.30 ± 2.57 | 50.46 ± 2.00 | 69.35 ± 1.65 | 51.72 ± 1.24 |
| DropEdge | 85.55 ± 6.07 | 72.86 ± 4.60 | 74.60 ± 2.72 | 49.93 ± 3.75 | 69.63 ± 1.11 | 53.40 ± 0.57 |
| DropNode | 85.02 ± 7.24 | 75.01 ± 4.99 | 76.10 ± 2.77 | 50.60 ± 2.52 | 70.01 ± 1.33 | 52.92 ± 0.82 |
| DropMessage | 85.60 ± 6.94 | 74.80 ± 4.52 | 75.50 ± 1.56 | 50.46 ± 1.88 | 69.82 ± 1.12 | 53.34 ± 0.76 |
| RADE-OF | 85.86 ± 8.13 | 75.68 ± 5.18 | **76.30 ± 2.42** | 50.64 ± 3.31 | 70.48 ± 1.22 | 53.90 ± 0.81 |
| RADE-OFS | **86.10 ± 7.19** | 75.82 ± 4.92 | 76.20 ± 2.85 | **50.78 ± 2.22** | **70.55 ± 1.09** | **54.48 ± 0.76** |

## H.1. Results

**Node Classification.** Table 7 reports node-classification results with GIN and GAT backbones. With GIN, RADE-OF is the strongest overall: it is best on six of the eight datasets (Cora, CiteSeer, CS, Physics, Flickr, and ogbn-arxiv) and second on Computer. PubMed is the main exception, where RADE-OFS is best, while RADE-OF remains close to the strongest non-RADE baseline. On the datasets where RADE-OF is best, its gains over the strongest non-RADE baseline range from 0.05% on Physics to 0.44% on Cora. RADE-OFS also improves over the GIN backbone on all datasets, but is typically below RADE-OF except on PubMed.

*Table 9.* Node-classification ablations of GradNorm (GN), Expectation-Preserving Correction (EPC), and nonlinearity for RADE-OF (mean±std over five runs) under GIN and GAT backbones. Accuracy (%) is reported on all datasets except Flickr with micro-F1 (%). RADE-OF-Lin and GIN-Lin have the linearized GIN backbone. Best is in bold.

| Variant | Cora | CiteSeer | PubMed | CS | Computer | Physics | Flickr | ogbn-arxiv |
|---|---|---|---|---|---|---|---|---|
| | | | | **Backbone: GIN** | | | | |
| RADE-OF | **$78.64_{\pm0.68}$** | **$65.94_{\pm1.45}$** | **$77.09_{\pm0.95}$** | **$93.01_{\pm0.14}$** | **$90.50_{\pm0.15}$** | **$96.25_{\pm0.10}$** | **$53.35_{\pm0.08}$** | **$71.85_{\pm0.32}$** |
| RADE-OF w/o GN | $78.24_{\pm2.26}$ | $63.74_{\pm1.87}$ | $76.60_{\pm0.68}$ | $92.73_{\pm0.25}$ | $89.72_{\pm0.33}$ | $96.24_{\pm0.04}$ | $52.74_{\pm0.31}$ | $71.04_{\pm0.14}$ |
| RADE-OF w/o GN & EPC | $78.20_{\pm0.78}$ | $63.60_{\pm2.07}$ | $76.46_{\pm0.79}$ | $92.42_{\pm0.44}$ | $88.94_{\pm0.71}$ | $96.22_{\pm0.10}$ | $52.55_{\pm0.23}$ | $70.90_{\pm0.35}$ |
| RADE-OF-Lin | $76.48_{\pm1.18}$ | $63.72_{\pm1.10}$ | $76.64_{\pm0.26}$ | $91.58_{\pm0.27}$ | $87.80_{\pm0.48}$ | $95.59_{\pm0.12}$ | $50.62_{\pm0.30}$ | $65.95_{\pm0.30}$ |
| RADE-OF-Lin w/o GN | $76.32_{\pm1.03}$ | $61.45_{\pm2.88}$ | $76.36_{\pm0.84}$ | $91.54_{\pm0.11}$ | $87.52_{\pm0.41}$ | $95.57_{\pm0.16}$ | $50.30_{\pm0.26}$ | $65.73_{\pm0.41}$ |
| RADE-OF-Lin w/o GN & EPC | $75.98_{\pm1.30}$ | $61.85_{\pm1.30}$ | $76.20_{\pm0.98}$ | $91.49_{\pm0.14}$ | $86.24_{\pm0.55}$ | $95.50_{\pm0.12}$ | $50.32_{\pm0.37}$ | $65.55_{\pm0.45}$ |
| GIN-Lin | $75.84_{\pm1.69}$ | $62.56_{\pm1.23}$ | $75.92_{\pm1.05}$ | $91.34_{\pm0.18}$ | $84.56_{\pm1.22}$ | $95.43_{\pm0.13}$ | $50.16_{\pm0.16}$ | $67.19_{\pm0.18}$ |
| | | | | **Backbone: GAT** | | | | |
| RADE-OF | **$80.32_{\pm0.78}$** | **$66.34_{\pm0.85}$** | **$76.96_{\pm0.60}$** | **$92.08_{\pm0.13}$** | **$90.40_{\pm0.36}$** | **$96.03_{\pm0.25}$** | $51.98_{\pm0.21}$ | **$70.98_{\pm0.12}$** |
| RADE-OF w/o GN | $79.46_{\pm1.56}$ | $65.62_{\pm2.15}$ | $76.78_{\pm0.98}$ | $88.52_{\pm3.00}$ | $90.27_{\pm0.16}$ | $95.95_{\pm0.16}$ | **$52.04_{\pm0.27}$** | $70.92_{\pm0.15}$ |
| RADE-OF w/o GN & EPC | $77.54_{\pm0.94}$ | $62.68_{\pm2.14}$ | $76.64_{\pm0.32}$ | $88.28_{\pm1.24}$ | $90.25_{\pm0.20}$ | $95.92_{\pm0.10}$ | $51.82_{\pm0.18}$ | $70.83_{\pm0.20}$ |

With GAT, RADE variants are best on seven of the eight datasets. RADE-OFS is best on PubMed, while RADE-OF is best on Cora, CS, Computer, Physics, Flickr, and ogbn-arxiv. CiteSeer is the only exception, where Dropout is best and RADE-OF is second. Both RADE variants improve over the GAT backbone, with RADE-OF again serving as the stronger default and RADE-OFS being most competitive on PubMed. These results show that the gains of RADE are not specific to GCN and extend to both sum-aggregation and attention-based backbones.

**Graph Classification.** Table 8 reports graph-classification results with GIN and GAT backbones. With GIN, both RADE variants improve over the backbone on all datasets. RADE-OFS is best on four of the six datasets (PROTEINS, IMDB-M, ogbg-molhiv, and peptides-func), while RADE-OF is best on MUTAG. The exception is IMDB-B, where DropMessage is best and RADE-OFS is second. The gains are largest on peptides-func, where RADE-OFS improves over the strongest non-RADE baseline by 0.64 AP, consistent with peptides-func requiring long-range interactions (Dwivedi et al., 2022).

With GAT, one of the RADE variants is best on every dataset. RADE-OFS is best on five datasets (MUTAG, PROTEINS, IMDB-M, ogbg-molhiv, and peptides-func), while RADE-OF is best on IMDB-B. Both RADE variants improve over the GAT backbone on all datasets. Overall, these graph-classification results support the same pattern observed with GCN: RADE-OF is a reliable regularizer, while RADE-OFS is especially effective when additional inference-time propagation benefits graph-level prediction.

## H.2. Ablation Studies under GIN and GAT Backbones

**GradNorm, Corrections, and Nonlinearity.** Table 9 studies the RADE-OF components under GIN and GAT backbones. We use *w/o GN* to denote removing GradNorm rate adaptation and *w/o GN & EPC* to denote removing both GradNorm and the expectation-preserving aggregation correction. Under GIN, removing GradNorm weakens RADE-OF on all datasets, with larger drops on CiteSeer, Computer, Flickr, and ogbn-arxiv. Removing both GradNorm and the expectation-preserving correction degrades performance more broadly, supporting the importance of controlling both perturbation strength and train-inference aggregation mismatch. The same pattern largely holds under GAT: removing GradNorm hurts RADE-OF on seven of eight datasets, with Flickr as the only small exception, while removing both GradNorm and the expectation-preserving correction is worse than full RADE-OF on all datasets.

For GIN, we also report RADE-OF-Lin, a linearized backbone that removes nonlinearities and matches the linear setting used in the variance-regularization derivation in Eq. (5). RADE-OF-Lin improves over the linear GIN baseline on seven of eight datasets, showing that RADE remains effective in the linearized setting. However, nonlinear RADE-OF is consistently stronger, with especially large gaps on CiteSeer, Flickr, and ogbn-arxiv. Removing GradNorm from RADE-OF-Lin also weakens performance on most datasets. Overall, these ablations show that adaptive rate selection and expectation-preserving correction remain useful beyond GCN, while nonlinear expressivity is important in practice. We do not report an analogous linearized GAT variant because, even without activation functions, GAT still uses representation-dependent softmax attention

*Table 10.* Drop-only vs. add-only decomposition of RADE-OF on node classification (mean±std over five runs) with GIN and GAT backbones. Accuracy (%) is reported on all datasets except Flickr, where we report micro-F1 (%). Best is in bold.

| Method | Cora | CiteSeer | PubMed | CS | Computer | Physics | Flickr | ogbn-arxiv |
|---|---|---|---|---|---|---|---|---|
| **Backbone: GIN** | | | | | | | | |
| RADE-OF | **78.64 ± 0.68** | **65.94 ± 1.45** | **77.09 ± 0.95** | **93.01 ± 0.14** | **90.50 ± 0.15** | **96.25 ± 0.10** | **53.35 ± 0.08** | **71.85 ± 0.32** |
| RADE-OF-Drop | 78.14 ± 0.94 | 64.82 ± 2.74 | 76.54 ± 1.06 | 91.98 ± 0.19 | 89.70 ± 1.12 | 95.98 ± 0.05 | 52.82 ± 0.12 | 71.09 ± 0.43 |
| RADE-OF-Add | 78.54 ± 1.10 | 64.40 ± 1.62 | 76.98 ± 0.85 | 92.31 ± 0.20 | 89.92 ± 0.51 | 96.02 ± 0.06 | 52.72 ± 0.19 | 71.08 ± 0.50 |
| **Backbone: GAT** | | | | | | | | |
| RADE-OF | **80.32 ± 0.78** | **66.34 ± 0.85** | **76.96 ± 0.60** | **92.08 ± 0.13** | **90.40 ± 0.36** | **96.03 ± 0.25** | **51.98 ± 0.21** | **70.98 ± 0.12** |
| RADE-OF-Drop | 79.54 ± 0.87 | 66.28 ± 1.19 | 76.58 ± 1.34 | 91.53 ± 0.78 | 90.12 ± 0.14 | 95.84 ± 0.09 | 51.93 ± 0.12 | 70.26 ± 0.32 |
| RADE-OF-Add | 79.14 ± 1.12 | 65.56 ± 1.49 | 76.72 ± 0.75 | 91.97 ± 0.28 | 90.37 ± 0.24 | 95.80 ± 0.12 | 51.84 ± 0.08 | 70.51 ± 0.26 |

and therefore does not reduce to a straightforward linear message-passing model.

**Decomposing RADE: Drop and Add.** Table 10 decomposes RADE-OF into drop-only and add-only components under GIN and GAT backbones. The full add-drop variant is best on all datasets for both backbones, improving over the better single-component variant by up to 1.12% under GIN (CiteSeer) and 0.78% under GAT (Cora). The better single component also varies across datasets and backbones: e.g., add-only is stronger on GIN PubMed, while drop-only is stronger on GIN CiteSeer. Thus, neither primitive consistently dominates the other, and combining deletion with addition gives the most reliable performance.

**Adaptive vs. Fine-tuned Rates.** Table 11 compares GradNorm-adaptive rate selection with fixed tuned rates under GIN and GAT backbones. Adaptive selection is stronger in 27 of the 32 method-backbone-dataset comparisons. Under GIN, RADE-OF improves over its tuned counterpart on all datasets, while RADE-OFS improves on six of eight, with small exceptions on Physics and Flickr. Under GAT, RADE-OF improves on six of eight datasets, except PubMed and Flickr, and RADE-OFS improves on seven of eight, except Computer. Overall, these results support GradNorm-based rate selection over using a single dataset-specific fixed pair $(p, q)$.

*Table 11.* Fixed fine-tuned vs. adaptive perturbation rates on node classification (mean±std over five runs) under GIN/GAT backbones. "Tuned" selects the best rate pair by hyperparameter search, while RADE-OF/RADE-OFS use adaptive GradNorm rule. Best is in bold.

| Method | Cora | CiteSeer | PubMed | CS | Computer | Physics | Flickr | ogbn-arxiv |
|---|---|---|---|---|---|---|---|---|
| **Backbone: GIN** | | | | | | | | |
| RADE-OF | **78.64 ± 0.68** | **65.94 ± 1.45** | **77.09 ± 0.95** | **93.01 ± 0.14** | **90.50 ± 0.15** | **96.25 ± 0.10** | **53.35 ± 0.08** | **71.85 ± 0.32** |
| RADE-OF (tuned) | 78.50 ± 0.42 | 64.84 ± 1.90 | 76.60 ± 0.68 | 92.85 ± 0.18 | 89.72 ± 0.33 | 96.24 ± 0.04 | 53.00 ± 0.16 | 71.40 ± 0.18 |
| RADE-OFS | **78.22 ± 2.17** | **65.69 ± 1.85** | **77.52 ± 1.19** | **92.75 ± 0.25** | **89.82 ± 0.48** | 96.24 ± 0.05 | 53.02 ± 0.15 | **70.85 ± 0.65** |
| RADE-OFS (tuned) | 78.18 ± 1.08 | 65.00 ± 1.42 | 77.04 ± 0.77 | 92.70 ± 0.21 | 89.05 ± 0.44 | **96.29 ± 0.05** | **53.05 ± 0.22** | 70.56 ± 0.47 |
| **Backbone: GAT** | | | | | | | | |
| RADE-OF | **80.32 ± 0.78** | **66.34 ± 0.85** | 76.96 ± 0.60 | **92.08 ± 0.13** | **90.40 ± 0.36** | **96.03 ± 0.25** | 51.98 ± 0.21 | **70.98 ± 0.12** |
| RADE-OF (tuned) | 80.06 ± 1.20 | 65.76 ± 1.96 | **76.98 ± 0.52** | 90.06 ± 0.18 | 90.27 ± 0.16 | 96.01 ± 0.16 | **52.06 ± 0.20** | 70.92 ± 0.15 |
| RADE-OFS | **80.22 ± 1.44** | **66.10 ± 0.72** | **77.02 ± 1.15** | **92.07 ± 0.24** | 90.24 ± 0.12 | **95.93 ± 0.14** | **51.94 ± 0.15** | **70.66 ± 0.16** |
| RADE-OFS (tuned) | 79.94 ± 0.68 | 65.96 ± 0.50 | 76.84 ± 0.91 | 91.90 ± 0.30 | **90.28 ± 0.15** | 95.89 ± 0.08 | 51.80 ± 0.10 | 70.52 ± 0.19 |

## H.3. Runtime Analysis

We report the runtime overhead of RADE-OF on node classification with the GCN backbone. Compared with GCN, RADE adds three sources of computation: sampling perturbed graphs, applying expectation-preserving aggregation corrections, and adapting $(p, q)$ with the GradNorm objective. Edge deletion is sampled over observed edges, while edge addition samples a fixed number of non-edges without constructing a dense complement. For GCN, the correction terms require degree-moment quantities under the perturbed graph. These are computed from node-wise degree statistics, cached graph statistics, global sums, and sparse scatter operations, rather than by enumerating all non-edges. Thus, RADE increases the

constant factor relative to GCN, but does not require dense $n \times n$ complement matrices.

Table 12 reports the average training time per epoch. RADE-OF w/o GN fixes the rates, so its gap from RADE-OF reflects the overhead of adaptive rate selection. RADE-OF w/o GN & EPC further removes the correction, so its gap from RADE-OF w/o GN reflects the overhead of expectation-preserving correction. The remaining runtime corresponds to the underlying stochastic add-drop sampling and training pipeline.

*Table 12.* Runtime analysis on node classification with the GCN backbone. We report mean±std training time per epoch in seconds. RADE-OF includes sampling, Expectation-Preserving Correction (EPC), and GradNorm (GN) rate adaptation.

| Method | Cora | CiteSeer | PubMed | CS | Computer | Physics | Flickr | ogbn-arxiv |
|---|---|---|---|---|---|---|---|---|
| RADE-OF | $0.103_{\pm0.035}$ | $0.111_{\pm0.036}$ | $0.295_{\pm0.036}$ | $0.195_{\pm0.043}$ | $0.460_{\pm0.035}$ | $0.423_{\pm0.030}$ | $10.263_{\pm0.888}$ | $18.986_{\pm1.489}$ |
| RADE-OF w/o GN | $0.051_{\pm0.032}$ | $0.050_{\pm0.034}$ | $0.159_{\pm0.030}$ | $0.146_{\pm0.014}$ | $0.288_{\pm0.012}$ | $0.290_{\pm0.017}$ | $8.770_{\pm0.265}$ | $15.400_{\pm0.463}$ |
| RADE-OF w/o GN & EPC | $0.042_{\pm0.033}$ | $0.046_{\pm0.034}$ | $0.139_{\pm0.030}$ | $0.138_{\pm0.012}$ | $0.263_{\pm0.019}$ | $0.277_{\pm0.035}$ | $8.394_{\pm0.225}$ | $14.561_{\pm0.589}$ |
| GCN | $0.018_{\pm0.033}$ | $0.025_{\pm0.038}$ | $0.052_{\pm0.031}$ | $0.035_{\pm0.015}$ | $0.068_{\pm0.011}$ | $0.079_{\pm0.015}$ | $4.577_{\pm0.141}$ | $7.143_{\pm0.358}$ |

The results show that stochastic add-drop sampling and GradNorm-based rate adaptation account for most of the added runtime. EPC adds a smaller cost: RADE-OF w/o GN remains close to RADE-OF w/o GN & EPC across datasets. Full RADE-OF is slower because it additionally computes the GradNorm objective and updates $(p, q)$ during training. Even with this cost, RADE-OF does not check or store all missing edges. It samples only the added edges it uses and computes correction terms from graph summary statistics. Thus, RADE-OF is slower than GCN, but avoids the much higher cost of dense all-pairs computation.

## I. Experimental Details

**Hardware Specification and Environment.** We ran all our experiments on a server with 40 CPU cores, 377 GB RAM and 11 GB GTX 1080 Ti GPU. The code is written in Python 3.8.0, PyTorch 2.0.0, and PyTorch Geometric 2.2.0.

**Dataset Statistics.** Table 13 reports the statistics of node- and graph-classification datasets.

**Dataset-Specific Hyperparameters.** We report dataset-specific backbone hyperparameters in Table 14 for node classification and Table 15 for graph classification, and the selected augmentation rates in Table 16 and Table 17. To ensure a controlled comparison, we use the same backbone configuration across all augmentation methods. For node classification, we select the hidden dimension from $\{64, 128, 256, 512\}$. For TU graph-classification datasets, we search over hidden dimensions $\{16, 32, 64\}$ and batch sizes $\{16, 32, 64\}$, and train with Adam using a StepLR scheduler. For ogbg-molhiv and peptides-func, we use the hidden dimensions and batch sizes reported in their respective benchmark protocols. For augmentation baselines, we tune the augmentation rate over $\{0.2, 0.3, 0.5, 0.7, 0.8, 0.9\}$. For graph classification, all models are trained with Adam, except peptides-func, for which we use AdamW with a ReduceLROnPlateau scheduler following the LRGB protocol (Dwivedi et al., 2022).

*Table 13.* Dataset statistics for node classification and graph classification.

| Node Classification | | | | | | | |
|---|---|---|---|---|---|---|---|
| **Dataset** | **#Nodes** | **#Edges** | **#Feats** | **#Classes** | **Train/Val/Test** | **Training** | **Metric** |
| Cora | $2,708$ | $5,429$ | $1,433$ | 7 | Official | Full-batch | Acc |
| CiteSeer | $3,327$ | $4,732$ | $3,703$ | 6 | Official | Full-batch | Acc |
| PubMed | $19,717$ | $44,338$ | $500$ | 3 | Official | Full-batch | Acc |
| Computer | $13,752$ | $245,861$ | $767$ | 10 | 60/20/20 | Full-batch | Acc |
| CS | $18,333$ | $81,894$ | $6,805$ | 15 | 60/20/20 | Full-batch | Acc |
| Physics | $34,493$ | $247,962$ | $8,415$ | 5 | 60/20/20 | Full-batch | Acc |
| Flickr | $89,250$ | $899,756$ | $500$ | 7 | Official | Mini-batch | Micro-F1 |
| ogbn-arxiv | $169,343$ | $1,166,243$ | $128$ | 40 | Official | Mini-batch | Acc |
| Graph Classification | | | | | | | |
| **Dataset** | **#Graphs** | **Avg #Nodes** | **Avg #Edges** | **#Feats** | **#Classes/Tasks** | **Train/Val/Test** | **Metric** |
| MUTAG | 188 | 17.9 | 39.6 | 7 | 2 | 10-fold CV | Acc |
| PROTEINS | 1,113 | 39.1 | 145.6 | 3 | 2 | 10-fold CV | Acc |
| IMDB-B | 1,000 | 19.8 | 193.1 | $65^*$ | 2 | 10-fold CV | Acc |
| IMDB-M | 1,500 | 13.0 | 65.9 | $59^*$ | 3 | 10-fold CV | Acc |
| ogbg-molhiv | 41,127 | 25.5 | 27.5 | 9 | 1 | Official | ROC-AUC |
| peptides-func | 15,535 | 150.94 | 307.30 | 9 | 10 | Official | AP |

\* IMDB-BINARY and IMDB-MULTI have no raw node attributes; we use one-hot degree features following the convention.

*Table 14.* Node classification backbone/training hyperparameters. Hidden dimensions can differ by backbone.

| **Dataset** | **Hidden channels** | | | **Training** | | **Batching** |
|---|---|---|---|---|---|---|
| | **GCN** | **GIN** | **GAT** | **LR** | **Epochs** | |
| Cora | 512 | 512 | 256 | 0.001 | 500 | Full |
| CiteSeer | 512 | 512 | 512 | 0.001 | 500 | Full |
| PubMed | 512 | 512 | 256 | 0.001 | 500 | Full |
| CS | 64 | 256 | 128 | 0.001 | 1500 | Full |
| Computer | 128 | 128 | 128 | 0.001 | 1000 | Full |
| Physics | 64 | 64 | 64 | 0.001 | 1500 | Full |
| Flickr | 256 | 256 | 256 | 0.001 | 1000 | Mini (2,048 nodes/batch) |
| ogbn-arxiv | 256 | 256 | 256 | 0.001 | 2000 | Mini (2,048 nodes/batch) |

*Table 15.* Graph classification backbone/training hyperparameters. Hidden dimensions and batch sizes can differ by backbone.

| **Dataset** | **Hidden channels** | | | **Training** | | **Batch size** | | |
|---|---|---|---|---|---|---|---|---|
| | **GCN** | **GIN** | **GAT** | **LR** | **Epochs** | **GCN** | **GIN** | **GAT** |
| MUTAG | 32 | 32 | 32 | 0.001 | 350 | 16 | 32 | 16 |
| PROTEINS | 64 | 32 | 64 | 0.001 | 350 | 32 | 32 | 16 |
| IMDB-B | 32 | 16 | 32 | 0.001 | 350 | 64 | 32 | 64 |
| IMDB-M | 32 | 16 | 32 | 0.001 | 350 | 64 | 32 | 64 |
| ogbg-molhiv | 300 | 300 | 300 | 0.001 | 350 | 256 | 256 | 32 |
| peptides-func | 300 | 300 | 300 | 0.001 | 350 | 128 | 128 | 128 |

*Table 16.* Node classification augmentation hyperparameters. The table summarizes the augmentation-specific rates used by each baseline. For RADE variants, we report the initialization $(p_0, q_0)$; GradNorm then adapts $(p, q)$ during training.

| Dataset | Dropout rate | DropEdge rate | DropNode rate | DropMessage rate | RADE $p_0$ | RADE $q_0$ |
|---|---|---|---|---|---|---|
| Cora | 0.70 | 0.50 | 0.90 | 0.50 | 0.50 | 0.000721 |
| CiteSeer | 0.50 | 0.20 | 0.90 | 0.90 | 0.50 | 0.000410 |
| PubMed | 0.70 | 0.20 | 0.20 | 0.90 | 0.50 | 0.000114 |
| CS | 0.30 | 0.20 | 0.50 | 0.90 | 0.50 | 0.000239 |
| Computer | 0.50 | 0.50 | 0.50 | 0.50 | 0.50 | 0.001300 |
| Physics | 0.30 | 0.20 | 0.20 | 0.20 | 0.50 | 0.000208 |
| Flickr | 0.50 | 0.20 | 0.20 | 0.50 | 0.50 | 0.000056 |
| ogbn-arxiv | 0.20 | 0.20 | 0.50 | 0.50 | 0.50 | 0.000040 |

*Table 17.* Graph classification augmentation hyperparameters. The table summarizes the augmentation-specific rates used by each baseline. For RADE variants, we report the initialization $(p_0, q_0)$; GradNorm then adapts $(p, q)$ during training.

| Dataset | Dropout rate | DropEdge rate | DropNode rate | DropMessage rate | RADE $p_0$ | RADE $q_0$ |
|---|---|---|---|---|---|---|
| MUTAG | 0.50 | 0.20 | 0.20 | 0.20 | 0.50 | 0.0605 |
| PROTEINS | 0.50 | 0.80 | 0.50 | 0.80 | 0.50 | 0.0203 |
| IMDB-B | 0.50 | 0.20 | 0.50 | 0.50 | 0.50 | 0.2044 |
| IMDB-M | 0.80 | 0.20 | 0.80 | 0.20 | 0.50 | 0.2884 |
| ogbg-molhiv | 0.50 | 0.20 | 0.20 | 0.50 | 0.50 | 0.0355 |
| peptides-func | 0.50 | 0.20 | 0.50 | 0.50 | 0.50 | 0.0052 |

