# OpenReview forum: "RADE: Random Add-Drop Edge as a Regularizer"
_ICML.cc/2026/Conference — ICML 2026 regular_

### Official Review · Reviewer_Qeus · 2026-02-21

**Soundness:** 4
**Presentation:** 4
**Significance:** 3
**Originality:** 3
**Overall Recommendation:** 6
**Confidence:** 4

**Summary:**

This paper proposes RADE, an expectation-preserving graph augmentation strategy that independently drops existing edges and adds non-edges. By maintaining the expected message aggregation at each layer, RADE eliminates train-test mismatch and acts as a variance-weighted logit-space regularizer. The authors introduce a GradNorm-based adaptive scheduling method to tune the drop/add rates dynamically during training. Additionally, an inference-correction variant (RADE-IC) is proposed to create test-time shortcuts that alleviate over-squashing in long-range tasks.

**Compliance With Llm Reviewing Policy:**

Affirmed.

**Final Justification:**

I appreciate the authors’ thoughtful rebuttal and the additional clarifications. The response clearly narrows the scope of the main claim, makes the practical role of the add-drop design more convincing, and improves transparency about the implementation trade-offs. Overall, the rebuttal increases my confidence in the paper’s framing and technical soundness. Therefore, I am comfortable raising my score.

**Key Questions For Authors:**

1. Could you explicitly quantify or discuss the expected bias introduced by non-linear activation functions (via Jensen's inequality)? Doesn't this non-linearity fundamentally invalidate the "unbiased" claim for the end-to-end model?
2. Could you provide a brief discussion (or empirical profiling) on the computational overhead introduced by calculating the mixed-binomial moments for the GCN-style normalization during the forward pass? Is this overhead strictly justified by the performance gains when scaling to large graphs?
3. Regarding the 'detached-variance convention' used for the GradNorm implementation (Appendix H): How much does this stop-gradient operation theoretically impact the optimality of the epoch-wise $(p,q)$ selection? Does the network effectively optimize a different objective than the one formally derived?
4. How do you address the concern that uniform random edge addition inherently introduces heterophilic noise and corrupts the underlying graph manifold?

**Limitations:**

Yes. The authors adequately discuss their limitations, explicitly noting the approximation gap introduced by non-linearities and transparently detailing the engineering compromises required for the GradNorm implementation.

**Strengths And Weaknesses:**

**Strengths:**
* **Mathematical Formalization & Intellectual Honesty:** The formalization of heuristic edge add/drop mechanisms into a mathematical framework is elegantly presented. Furthermore, the paper is exceptionally well-written, and the authors demonstrate commendable intellectual honesty by transparently discussing certain theoretical gaps rather than hiding them.
* **Practicality:** The GradNorm-based adaptive tuning provides a practical engineering solution that effectively mitigates the need for exhaustive, dataset-specific hyperparameter tuning for drop/add rates.

**Weaknesses:**
* **Theoretical Breakdown in Non-linear Deep Networks:** While the "Unbiased" claim holds under linear expectations within a single aggregation step, it fundamentally breaks down in practical deep GNNs due to non-linear activation functions (e.g., ReLU). By Jensen's inequality, $\mathbb{E}[f(x)] \neq f(\mathbb{E}[x])$, meaning the entire forward pass is inherently biased. Presenting the framework as "Unbiased" in the context of deep learning is somewhat of a misnomer.
* **Physical Intuition of "Random Add":** While dropping edges acts as a standard dropout regularizer, uniformly adding random edges fundamentally destroys the topological manifold and homophily of real-world sparse graphs. It injects severe heterophilic noise, which might explain the somewhat modest empirical gains.
* **Complexity vs. Gain:** The mathematical and computational overhead for the GCN-style symmetric normalization appears excessively high, requiring complex delta-method mixed-binomial approximations, yielding only marginal empirical improvements over standard DropEdge.
* **Theoretical Compromise in Optimization:** The reliance on a "detached-variance convention" (applying stop-gradient) for the GradNorm implementation is a necessary engineering trick, but it severs the exact equivalence between the theoretical objective and the actual optimized gradients.

---

> ### Author Rebuttal · Authors · 2026-03-31
>
> We thank the reviewer for the thoughtful comment and the positive assessment. We address the main concerns below and clarify them in the revision.
>
> **Theoretical Breakdown in Non-linear Deep Networks**
>
> We agree that in nonlinear GNNs, the end-to-end forward pass is not exactly unbiased. Our guarantee is thus narrower: RADE is expectation-preserving at the aggregation level, while the variance-based regularization view is a logit-level approximation built on top of that property. We noted this distinction in the paper, but we agree that the term “Unbiased” could be read too broadly. In revision, we emphasize expectation-preserving aggregation more explicitly and reorganize the method around that notion. We also include the RADE-Lin ablation to separate the “exact linear case” from the practical nonlinear setting: RADE-Lin makes the logit-level assumption exact, but the full nonlinear RADE is usually preferable empirically because preserving nonlinear expressivity matters in practice.
>
> **Physical Intuition of Random Edge Addition**
>
> Uniform random edge addition by itself can be harmful, especially if viewed as a standalone way to improve the original graph structure. Our claim is not that random addition alone is a strong default strategy. In fact, our ablations point in the opposite direction: add-only is generally weaker than full RADE, and on datasets such as CiteSeer, it is notably worse. The empirical evidence instead supports a different point: edge deletion and edge addition affect message passing differently, and their combination is stronger than either component alone. This is why we emphasize add-drop complementarity, rather than presenting random addition by itself as the main contribution.
>
> **Complexity vs. Gain**
>
> Although GCN instantiation is the heaviest version, the additional complexity does not come from enumerating non-edges or performing dense computations over the graph complement. In our implementation, the required complement-based quantities are obtained from cached input-graph statistics, global sums, and sparse scatter operations, while the delta-method corrections are inexpensive closed-form node-wise approximations. Thus, the practical overhead remains linear in graph size, with a higher constant factor rather than combinatorial or quadratic growth.
>
> **Theoretical Compromise in Optimization**
>
> We agree that this is a theoretical compromise. With the detached-variance convention, the GradNorm step is no longer the exact gradient-based optimization of the full theoretical objective in (p,q), but a practical heuristic for selecting the rates in each epoch. This approximation affects only the adaptive tuning of (p,q), not the RADE mechanism or its aggregation-level correction. We chose it to keep the tuning step tractable in practice, and we will state this limitation more explicitly in the revision.
>
> **Questions**
>
> **Q1: Bias from Nonlinearities.** In general, this bias depends on the specific architecture, activations, normalization layers, and mask-sharing scheme, so we do not believe there is a simple universal closed-form quantity that can be stated without adding substantial new analysis. For this reason, we do not claim an explicit end-to-end bias bound in the current paper. Deriving architecture-dependent bounds on the additional bias introduced by nonlinearities is an interesting direction for future work, and we will note this in the revision.
>
> **Q2: Forward-Pass Runtime Overhead.** For the discussion, please see our response to the **Complexity vs. Gain**. To make this overhead more transparent, we provide a time-per-epoch analysis (in seconds, averaged over 5 runs) below. The results show that the correction increases runtime slightly, and remains practically manageable relative to the performance gains. We will include a more comprehensive runtime analysis in the revised paper.
>
> | Dataset | RADE w/o Correction | RADE with Correction |
> |---|---:|---:|
> | Cora | 0.0447s ± 0.0012s | 0.0479s ± 0.0026s |
> | CiteSeer | 0.0315s ± 0.0005s | 0.0401s ± 0.0013s |
> | Pubmed | 0.1056s ± 0.0053s | 0.1122s ± 0.0012s |
> | Physics | 0.1949s ± 0.0008s | 0.2116s ± 0.0005s |
> | Computer | 0.2557s ± 0.0024s | 0.2872s ± 0.0004s |
> | CS | 0.1228s ± 0.0010s | 0.1233s ± 0.0012s |
> | Flickr | 6.3088s ± 0.1452s | 6.3248s ± 0.0522s |
> | ogbn-arxiv | 10.1171s ± 0.1447s | 11.0399s ± 0.0260s |
>
> **Q3: Effect of the Stop-Gradient Convention.** Please see **Theoretical Compromise in Optimization paragraph**.
>
> **Q4: Concern About Heterophilic Noise.** As discussed above, we do not rely on add-only. RADE uses edge addition only as part of a controlled add-drop combination, and our ablations show that this combination is stronger than either primitive alone. We will clarify this in the revision.

---

> > ### Author Rebuttal · Reviewer_Qeus · 2026-04-01
> >
> > I appreciate the authors’ thoughtful rebuttal and the additional clarifications. The response clearly narrows the scope of the main claim, makes the practical role of the add-drop design more convincing, and improves transparency about the implementation trade-offs. Overall, the rebuttal increases my confidence in the paper’s framing and technical soundness. Therefore, I am comfortable raising my score.

---

### Official Review · Reviewer_rdhW · 2026-03-11

**Soundness:** 2
**Presentation:** 2
**Significance:** 2
**Originality:** 2
**Overall Recommendation:** 3
**Confidence:** 4

**Summary:**

This paper studies regularization for graph neural networks, with emphasis on mitigating overfitting and alleviating over-squashing through additional inference-time message pathways. The method randomly drops existing edges and adds sampled non-edges while correcting the aggregation so that the expected message aggregation matches that of the clean graph. The paper develops a logit-space view in which unbiased perturbations induce a variance-weighted regularization term. Experiments compare RADE and RADE-IC against common stochastic regularization baselines on multiple benchmark datasets.

**Compliance With Llm Reviewing Policy:**

Affirmed.

**Key Questions For Authors:**

The motivation regarding the limitations of existing perturbation approaches is insufficiently justified. In the introduction, the manuscript claims that current augmentation strategies overlook four aspects. However, the text does not provide detailed explanations or supporting evidence for these claims, making the argument less convincing in its current form.

The related work discussion lacks coverage of the most recent literature. The manuscript reviews several earlier studies on graph augmentation and topology perturbation, but the discussion does not sufficiently incorporate more recent works published around 2025. A more comprehensive review of the latest developments would help position the contribution relative to the current state of the field.

Some important concepts and notation in the theoretical sections are not clearly defined. In particular, the meaning of some parameters and the notion of perturbation strength are not formally introduced before being used in the analysis surrounding Propositions 4.2 and 5.1. This lack of precise definition makes the theoretical derivations more difficult to follow.

The presentation of the proposed method lacks a clear procedural overview. After reading the method section, the overall pipeline of RADE remains difficult to reconstruct, as the interactions among edge dropping, non-edge sampling, unbiased correction, and adaptive rate selection are described across multiple sections without a concise summary. A framework diagram or algorithmic description would help clarify the workflow.

The evaluation and related-work discussion do not sufficiently acknowledge prior studies addressing similar problems through graph perturbation. Existing research areas such as differential privacy on graph data, fairness-oriented structural perturbation, and graph structure learning also modify graph topology to address similar challenges. These lines of work should be more thoroughly discussed and, where relevant, included in experimental comparisons to better contextualize the proposed method.

**Limitations:**

refer to questions

**Strengths And Weaknesses:**

The formulation of expectation-preserving graph perturbation is clearly motivated, and the method is presented as more than a heuristic by connecting topology perturbations to a variance-based regularization view in logit space. The decomposition of drop-induced and add-induced variance provides a useful conceptual lens for understanding why the two operations may be complementary rather than interchangeable.

---

> ### Author Rebuttal · Authors · 2026-03-31
>
> We thank the reviewer for the careful reading and for recognizing the motivation behind our work. We address the reviewers' concerns below and clarify them in the revision.
>
> **Motivation in the Introduction**
>
> The submitted introduction stated these limitations too compactly and did not provide enough immediate justification for each one. Our intention was not to suggest that every existing perturbation method suffers equally from all four issues, but rather to summarize the specific practical gaps that motivate our formulation. In the revision, we reworked the introduction to make these points more precise. Rather than presenting a broad list of unsupported shortcomings, we now separate the discussion by augmentation purpose: for overfitting, we highlight tuning sensitivity and train-inference aggregation mismatch; for over-squashing, we explain that rewiring and shortcut-based methods improve long-range communication but are typically framed differently from stochastic regularization. We then motivate RADE as a unified framework bridging these two lines of work.
>
> **Coverage of Recent Related Work**
>
> In the revision, we have expanded and reorganized the discussion by augmentation purpose and added more recent work on both stochastic regularization and topology modification so that the contribution is positioned more clearly relative to the current state of the field. We appreciate this comment and will continue to strengthen the coverage of recent literature in the final version. If there are specific recent papers the reviewer believes are particularly relevant, we would be grateful to consider them as well.
>
> **Definitions and Theoretical Clarity**
>
> In the revision, we made the theoretical sections more explicit and easier to follow. We now introduce the perturbation process and its parameters earlier, define the corrected aggregation rules explicitly for the two variants, and make the role of perturbation strength precise through the variance-based regularization view. The revised draft explains perturbation strength in terms of how the deletion and addition rates control the variance of the induced logit perturbation, and then uses that definition to motivate the adaptive control section.
>
> **Procedural Overview of RADE**
>
> We have improved this by opening the method section with a clearer two-part view of RADE: how the perturbed graph is sampled, and how aggregation is corrected on the sampled view. We also make the distinction between the two variants more explicit: RADE preserves the input-graph aggregation in expectation, whereas RADE-IC aligns training with a modified inference aggregation that retains the effect of edge addition for long-range communication.
> To make the procedure even easier to follow, we will add a concise algorithmic summary in the final revision: sample dropped/added edges using (p,q) → apply the appropriate corrected aggregation rule (RADE or RADE-IC) → train on the sampled view → update (p,q) at each epoch via the GradNorm-based rule.
>
> **Scope of Empirical Comparisons**
>
> Our primary focus is on graph perturbation for two specific objectives: (i) regularization against overfitting, and (ii) improving long-range communication to mitigate over-squashing. In the revision, we made this scope explicit and expanded the related work to acknowledge that graph topology is also modified for other objectives beyond the two that are central to our study. In particular, the revised related work now includes a separate discussion of topology perturbations used for other purposes.
> At the same time, we would like to clarify why we did not include all such methods in the empirical comparison. Areas such as differential privacy, fairness-oriented perturbation, and graph structure learning also modify graph topology, but they are typically motivated by different objectives and evaluated under different criteria from ours. In our experiments, our direct comparisons focus on perturbation methods that are closest in purpose to ours: stochastic regularizers for overfitting and topology-modification methods aimed at improving message passing. The revised paper now makes this scope and positioning clearer.
> We will make the boundary of our comparison set more explicit in the final version. If the reviewer has specific works in mind from these areas that they believe are especially relevant to RADE, we would be grateful to examine and cite them as well.

---

> > ### Author Rebuttal · Reviewer_rdhW · 2026-04-03
> >
> > Thank you for the rebuttal. The paper would benefit from further improvements in rigor, particularly in the use of symbols, clarity of statements, depth of discussion, and overall writing quality. While some concerns (e.g., W1, W4) have been partially addressed, additional refinement is still needed. I encourage the authors to address these aspects comprehensively and further polish the manuscript in the next revision.

---

### Official Review · Reviewer_Kyrb · 2026-03-11

**Soundness:** 3
**Presentation:** 3
**Significance:** 2
**Originality:** 2
**Overall Recommendation:** 4
**Confidence:** 3

**Summary:**

This paper proposes **Unbiased Random Add-Drop Edge (RADE)**, a graph augmentation method that randomly flips entries in the adjacency matrix while preserving the expectation of aggregated node features through a specially designed unbiased aggregation strategy. The paper provides a theoretical analysis of the method and evaluates it against several existing graph augmentation baselines. Empirically, RADE shows modest improvements over prior methods on some benchmarks.

**Compliance With Llm Reviewing Policy:**

Affirmed.

**Final Justification:**

After reviewing the rebuttal and considering the other reviews, I acknowledge the author's contribution in demonstrating the "non-interchangeability" of drop-only and add-only strategies, as well as the empirical evidence showing that full RADE exhibits greater expressiveness than RADE-Drop and RADE-Add.

Based on this, I decide to increase my score to 4.

**Key Questions For Authors:**

Please see the weaknesses.

**Limitations:**

Introducing stochastic edge perturbations may increase training instability and make the model more difficult to converge.

**Strengths And Weaknesses:**

**Strengths**

* The proposed method is simple and easy to understand. Its design motivation is also clear: random edge perturbation is combined with an unbiased aggregation correction to reduce train-test mismatch.
* The ablation studies are relatively comprehensive and help verify the effectiveness of the proposed components.

**Weaknesses**

* **The overall contribution appears limited.** On the theory side, the paper mainly builds on the augmentation perspective introduced in prior work such as *DropMessage*. Much of the paper is devoted to deriving propositions and corollaries under this existing framework, but it is unclear what the main new theoretical result is beyond analyzing a particular instantiation. The paper provides a theoretical justification for the method, but I do not see a sufficiently strong new theorem or conceptual advance. Besides, the core method is essentially a random edge addition/deletion scheme with a correction mechanism to maintain unbiased aggregation. As also acknowledged in the paper, random edge dropping and adding have already been widely explored in prior graph augmentation literature. Therefore, the method seems more like a careful refinement of existing ideas than a substantially new approach.
* **The empirical gains are not very convincing.** In Table 1, the performance differences between the proposed method and competing methods appear quite small. While RADE is sometimes better, the margins do not seem large enough to clearly demonstrate a strong practical advantage over existing graph augmentation approaches.

---

> ### Author Rebuttal · Authors · 2026-03-31
>
> We thank the reviewer for their thoughtful comments. We address their major concerns below and will revise the paper accordingly.
>
> **Overall Contribution**
>
> Regarding the concern that the overall contribution is limited, we agree that random edge deletion and random edge addition, each as an individual perturbation method, are not novel, but the novelty lies in their combinations. This combination, in a principled stochastic framework with train-inference alignment, unlocks unifying two uses of graph perturbation that prior work has mostly treated separately: stochastic regularization for overfitting and connectivity enrichment (known as rewiring) for mitigating over-squashing. On the theory side, we agree that theoretical results per se are not the main strength of the paper, and our proof techniques are not necessarily novel or substantial. However, theoretical analyses served well our main goal of unifying perturbation techniques for two purposes of over-squashing and overfitting through principled stochastic perturbations. They specifically allow us to derive train-inference alignment corrections for any variants of RADE on any backbone model with weighted-sum aggregation rules (GCN, GIN, etc.). Further, for the first-time, we show and characterize that drop-only and add-only are generally complementary rather than interchangeable (which is a formal reason why combining deletion and addition can yield benefits that neither primitive alone captures). Most importantly, our analyses fuelled the design of our adaptive procedure for selecting deletion and addition rates via GradNorm-style balancing. This technique avoids dataset-specific hyper-parameter tuning, rendering our augmentation methods practically parameter-free, while outperforming its fine-tune alternative due to their superior adaptive nature (as witnessed in our empirical studies).
>
> **Empirical Gains in Table 1**
>
> We agree that the gains in Table 1 are modest in absolute terms. Node classification benchmarks are less prone to over-squashing, so our method primarily acts only as a regularizer to reduce overfitting. Thus, one should not expect substantial gains from regularization alone. Note that RADE remains a consistently strong baseline under the same backbone and training pipeline, while also avoiding dataset-specific tuning of the add/drop rates through its adaptive scheduling. The stronger effect appears when RADE variants can address both overfitting and over-squashing. This is why the gains are more pronounced in the graph-classification results, where RADE-IC can address both overfitting and over-squashing. In those settings, the joint variant achieves first- or second-place performance and improves by 2.79% on MUTAG and 3.24% on peptides-func. We will revise the discussion around Table 1 to make this distinction more explicit: on the node-classification benchmarks, RADE mainly provides a regularization benefit, which naturally leads to modest but consistent gains; on datasets where long-range interaction matters more, the stronger advantages of the joint variant become clearer.
>
> **Training Instability**
>
> We agree that stochastic edge perturbation can, in principle, increase optimization noise. In RADE, however, the perturbation is paired with expectation-preserving aggregation correction, which avoids a systematic shift in the expected aggregation under perturbation, and its strength is also controlled adaptively through the GradNorm-style procedure. This helps prevent overly weak or overly strong perturbations during training that may cause instability. Empirically, we did not observe instability severe enough to prevent effective training on the reported benchmarks. We will clarify this point in the revision.

---

> > ### Author Rebuttal · Reviewer_Kyrb · 2026-04-01
> >
> > Thank you for your rebuttal. It helps clarify several of my concerns, and I appreciate the additional explanation. That said, I still feel that some points would benefit from further justification.
> >
> > First, regarding the claimed contribution, the authors state that they are the first to show and characterize that drop-only and add-only strategies are generally complementary. At the moment, I am not fully convinced that the current evidence is sufficient to support this claim. My understanding is that this argument mainly relies on Corollary 5.3. However, Corollary 5.3 appears to show only that the variance can be decomposed into two terms, corresponding to neighbors and non-neighbors. While this is an interesting observation, it does not by itself seem strong enough to establish “complementarity” in a broader sense. To make this point more convincing, it would be helpful to provide stronger theoretical evidence showing that jointly using both drop and add operations can realize variance patterns that neither single strategy can achieve on its own.
> >
> > Second, regarding the reported performance gains, the rebuttal attributes the improvement to task properties and highlights that, on graph tasks, “the joint variant achieves first- or second-place performance and improves by 2.79% on MUTAG and 3.24% on Peptides-func.” I appreciate this clarification, but I still find the empirical evidence somewhat mixed. In particular, under the GIN backbone, the proposed method shows a noticeable drop on MUTAG relative to DropNode in Table 7. Taken together, the current results seem suggestive, but in my view they are not yet sufficient to fully support a strong claim of consistent or significant performance improvement for the proposed method.
> >
> >
> > -----Edit------
> >
> > Thanks for the further clarification. Regarding the "complementary" claim, I find the term "non-interchangeability" acceptable and acknowledge that the overall better performance of full RADE compared to RADE-Drop and RADE-Add provides strong empirical evidence that jointly using both drop and add operations offers greater expressiveness. Based on the current situation, I decide to increase my score to 4.

---

> > > ### Author Response · Authors · 2026-04-03
> > >
> > > Thank you for this clarification. Our intended “complementary” claim relies mainly on Proposition 5.4, which is derived from Corollary 5.3. Corollary 5.3 shows that drop and add act on different node-wise statistics with disjoint support (neighbors vs. non-neighbors), while Proposition 5.4 gives a restrictive subsumption condition: under sum aggregation, exact matching of DROP-only and ADD-only across all nodes is possible only under a restrictive proportionality condition between these two statistics. Since this condition is generally not expected to hold, the two methods are not generally interchangeable. We agree, however, that the current theory does not yet establish the stronger claim that joint add-and-drop can realize variance patterns unattainable by either pure strategy alone. We also note that the current proposition statement and surrounding discussion should be clarified to emphasize this point more clearly (thanks for pointing this out to us). We will restate Proposition 5.4 to emphasize the non-interchangeability of add-only and drop-only by showing that they can be interchangeable under restrictive conditions. We will also revise the introduction to use the term non-interchangeable rather than complementary for these theoretical results. However, our empirical ablations are consistent with our complementary intuition, in that full RADE outperforms RADE-Drop and RADE-Add, but we agree that they do not replace the stronger theoretical characterization you are asking for, and we will revise the wording accordingly to avoid overclaiming.
> > >
> > > Regarding the empirical results, we agree that the empirical claim should be stated more cautiously, and that GIN/MUTAG is a counterexample to any stronger statement of universal improvement over all baselines. Our intended claim is thus not that RADE or RADE-IC dominates every regularizer on every benchmark, but that the method family is competitive overall, with its clearest gains appearing in settings where over-squashing is more pronounced. This is consistent with the results: on the GCN node-classification benchmarks, the RADE family attains the top result on 6/8 datasets, although with mostly modest margins; on GIN graph classification, RADE or RADE-IC is best on the other 5 out of 6 benchmarks (except MUTAG), including larger improvements on peptides-func and ogbg-molhiv. We also note that such dataset- and backbone-specific variation is not unusual in the random perturbation literature; for example, DropMessage explicitly notes that finding a universal method across datasets and models is challenging. We will revise the wording accordingly and avoid stronger phrases such as “consistent” or “significant improvement.”

---

### Official Review · Reviewer_sBzV · 2026-03-13

**Soundness:** 4
**Presentation:** 4
**Significance:** 3
**Originality:** 4
**Overall Recommendation:** 6
**Confidence:** 4

**Summary:**

The authors propose RADE and RADE-IC, which are two principled approaches to add or drop edges randomly during training and testing while ensuring that aggregations across layers are preserved in expectation. The former perturbs graph structure by randomly flipping edges through noise, while the latter also updates edges for aggregation in testing.

**Compliance With Llm Reviewing Policy:**

Affirmed.

**Final Justification:**

I maintain my original positive view of this work. The discussion with the authors addressed any of my concerns or questions. Their answers to my comments were nuanced, detailed, and satisfactory in content, as well as being concise given the space limitations.

**Key Questions For Authors:**

**Main questions:**
1. Line 091, column 2: "...these methods do not characterize the regularization effect..." How do they perform? Is this the only downside? This is not necessarily an insufficient gap, but perhaps you might want to emphasize why this warrants your approach.
2. Section 6: Both RADE and RADE-IC are well founded and explained clearly. However, by this point in the manuscript, I don't know if I completely understood the intuition behind when I should use RADE versus RADE-IC. Is RADE-IC preferable if you expect long-range interactions to be important, say for heterophilic data, whereas RADE is for more robust adjustments during training to help with overall generalization?
3. If CiteSeer is the main exception where DropNode is best in Table 1, do you have any intuition about this? As for RADE-IC performing best only on Pubmed, is over-squashing a known issue for Pubmed?
4. For your results in Table 3, while the comparison is thorough, it would be valuable to discuss why the results on Cora and CiteSeer align with your intuition, as this is the most useful part of the ablation study, as well as the most unique as it is specific to your method.

**Minor comments**
5. Line 123, column 1: You use $m$ to define the number of edges $|E|$, but later, you use $| \bar{E} |$ for the number of edges in the perturbed graph. It does not seem that $m$ appears again, and you also use lower-case $\mathbf{m}$ for node messages, albeit boldfaced. Perhaps you do not need $m$ for edges.

6. Perhaps you can introduce the aggregation coefficients $\Gamma_{ij}$ later, when you introduce the concept. The meaning of "aggregation coefficients" in your notation section may be confusing without context.

7. $\sigma(\cdot)$ is used in Proposition 4.1, but you have not explicitly defined it. I know that it is used in Appendix A, but in the body of the paper, the reader has not seen what this means.

8. Lines 189-190, column 1: I believe I know what you intended with "with entry $\delta_{i,c}$", but you may want to reword it.

9. You use $C$ to represent the number of classes as well as sets of non-neighbors $C(i)$. Perhaps you can change one or the other.

10. There is a comma instead of a period after the equation on line 689.

11. For the references, check incorrect capitalization and inconsistent venue formatting.

**Limitations:**

Yes

**Strengths And Weaknesses:**

This paper is clean, thorough, very clear, and provides a straightforward idea that is also sound and well justified. I share a few comments to address, which I think you are more than capable of addressing well within the rebuttal period. Your methodology section is far above average in terms of writing, clarity, and presentation. It is very thorough and nuanced, yet the idea is straightforward and your descriptions remain intuitive.

---

> ### Author Rebuttal · Authors · 2026-03-31
>
> We sincerely thank the reviewer for the positive assessment and for the thoughtful and constructive feedback. We address the reviewer’s questions below and have incorporated the corresponding clarifications in the revised version.
>
> **Question 1:** Many hybrid add-drop and rewiring approaches are effective for their intended goals (e.g.,  improving connectivity, robustness, or denoising), but are typically not formulated as stochastic regularizers with explicit train-inference alignment, nor provide a loss-level characterization of the regularization effect. The goal of our work differs in few ways when we drop and add edges: (i) regularization against overfitting, and (ii) simultaneously improved long-range interaction for mitigating over-squashing. To achieve these goals, we provide a principled way of train-inference alignment (through expectation-preserving aggregation) and, for the second goal, an inference-time correction that preserves the effect of added edges that mitigates over-squashing.
> In the revised paper, we clarify these distinctions by organizing augmentation strategies and literature  by their purpose and emphasizing that our principled stochastic framework addresses both overfitting and over-squashing in a unified way for the first time.
>
> **Question 2:** We will improve the intended use more clearly in revisions. The practical rule is: use RADE (that will be renamed to RADE-OF with OF for over-fitting) when the goal is regularization against overfitting; use RADE-IC (rename to RADE-OFS with additional S for squashing) when both regularization and long-range interaction (for oversquashing) need to be addressed.
>
> **Question 3:** Our intuition is that CiteSeer does not benefit from edge addition as much as the other node-classification benchmarks. This is consistent with our ablations, where add-only performs noticeably worse on CiteSeer, while drop-only remains much closer to full RADE. A second likely reason is that, for the GCN backbone, the correction relies on a delta-method approximation, which becomes less reliable when more low-degree nodes are present. CiteSeer is the only dataset with many isolated nodes and substantially more degree-1 nodes than the others, which likely makes the GCN correction less reliable on this dataset. We will revise the discussion to make this limitation on CiteSeer more explicit.
> For pubmed, after the reviewer's question, we measured its oversquashing metrics based on the methodology of (Saber and Salehi-Abari, 2025). Our results suggest that PubMed has a higher amount of over-squashing than most of the node-classification datasets considered in our studies (intensity = 0.13, moderate, and extremity = 0.29, extreme). This result helps explain why RADE-IC, designed to overcome oversquashing, is particularly effective on PubMed. We will add a brief discussion of it in the revision.
>
> **Question 4:** We agree that this should be explained more explicitly. The results on Cora and CiteSeer can be understood through two approximation effects in our method. For GCN, the RADE correction uses a delta-method approximation because it relies on degree-normalized random coefficients (as mentioned in footnote 4). This approximation is less accurate when many low-degree nodes are present. This likely explains the weaker behavior on CiteSeer, which has 48 isolated nodes and many more degree-1 nodes than the other datasets. This issue does not arise for GIN (Table 8), where the correction is exact, which is consistent with the stronger behavior of the corrected variants there. In addition, RADE-Lin makes the mean logit perturbation exactly zero, removing another approximation in the analysis, which helps explain why it is especially competitive on Cora and CiteSeer. On the remaining datasets, however, preserving nonlinear expressivity becomes more important, so the full RADE design is usually stronger. We will add this interpretation more explicitly in the revision.
>
> **Minor comments:** We thank the reviewer for these helpful comments. We have addressed all of them in the revision by adding missing definitions (e.g., the sigmoid function), reducing overloaded notation, rephrasing unclear sentences, and correcting punctuation and reference formatting.

---

> > ### Author Rebuttal · Reviewer_sBzV · 2026-04-02
> >
> > Thank you for your answers; I found your responses to me as well as to other reviewers to be thorough and well explained. I especially appreciated looking into how oversquashing can be a challenge for PubMed. I think it is very interesting. It points to potential ways to evaluate existing benchmark graph datasets based on how well RADE-OF or RADE-OFS perform. I have no more questions.

---

### Decision · Program_Chairs · 2026-04-30

**Decision:**

Accept (regular)

**Comment:**

The authors address overfitting and over-squashing in graph neural networks by proposing RADE and RADE-IC, which are two principled approaches to add or drop edges randomly during training and testing while ensuring that aggregations across layers are preserved in expectation. The paper provides theoretical analysis of the method and evaluate against several existing graph augmentation baselines to empirically show modest improvements over prior methods on some benchmarks.

Although the reviewers' opinions initially varied, they found the method to be clean and theoretically sound, and the ablation studies to be relatively comprehensive. The initial concerns were on the novelty of introducing random edge modifications, the suitability of some analysis only for linear models, and modest empirical performance. However, most reviewers became more convinced after the rebuttal and further discussion, with the general opinion that some of the contributions in the work (such as the "non-interchangeability" of drop-only and add-only strategies) are meaningful to the community, and that the overall work is solid. Hence, most reviewers favored acceptance of the paper.